# Translation in amino-acid-poor environments is limited by tRNA$^{Gln}$ charging

Natalya N Pavlova[1], Bryan King[1†], Rachel H Josselsohn[1†‡], Sara Violante[2], Victoria L Macera[1], Santosha A Vardhana[1], Justin R Cross[2], Craig B Thompson[1*]

[1]Cancer Biology & Genetics Program, Memorial Sloan Kettering Cancer Center, New York, United States; [2]The Donald B. and Catherine C. Marron Cancer Metabolism Center, Memorial Sloan Kettering Cancer Center, New York, United States

*For correspondence:
thompsonc@mskcc.org

†These authors contributed equally to this work

Present address: ‡Donald and Barbara Zucker School of Medicine at Hofstra University and Northwell Health, Hempstead, United States

**Abstract** An inadequate supply of amino acids leads to accumulation of uncharged tRNAs, which can bind and activate GCN2 kinase to reduce translation. Here, we show that glutamine-specific tRNAs selectively become uncharged when extracellular amino acid availability is compromised. In contrast, all other tRNAs retain charging of their cognate amino acids in a manner that is dependent upon intact lysosomal function. In addition to GCN2 activation and reduced total translation, the reduced charging of tRNA$^{Gln}$ in amino-acid-deprived cells also leads to specific depletion of proteins containing polyglutamine tracts including core-binding factor α1, mediator subunit 12, transcriptional coactivator CBP and TATA-box binding protein. Treating amino-acid-deprived cells with exogenous glutamine or glutaminase inhibitors restores tRNA$^{Gln}$ charging and the levels of polyglutamine-containing proteins. Together, these results demonstrate that the activation of GCN2 and the translation of polyglutamine-encoding transcripts serve as key sensors of glutamine availability in mammalian cells.

## Introduction

To be utilized in protein synthesis, amino acids must be first covalently attached, or charged, onto corresponding tRNA isoacceptors by amino-acid-specific tRNA synthetases. Imbalances in tRNA pools compromise translational fidelity and trigger proteotoxic stress (*Bloom-Ackermann et al., 2014*; *Nedialkova and Leidel, 2015*); thus, cells must continuously balance translational load with the extracellular amino acid availability. Eukaryotic cells possess sensors for monitoring both free and tRNA-charged amino acid reserves (reviewed in *González and Hall, 2017*). The former is carried out by a set of sensors that sample individual amino acids - namely, leucine, arginine, and S-adenosylmethionine (a derivative of methionine), and relay the amino acid sufficiency information to the mTORC1 complex (*Chantranupong et al., 2016*; *Gu et al., 2017*; *Wolfson et al., 2016*), whereas the latter is mediated by the GCN2 kinase, which becomes activated via autophosphorylation in response to uncharged tRNA binding (*Dong et al., 2000*).

In response to amino acid deprivation, inactivation of mTORC1 reduces cellular translational load while increasing the delivery of extracellular and intracellular protein substrates to the lysosome, where their proteolytic degradation provides a cell with an alternative supply of free amino acids. Consequently, even though an interruption in amino acid supply initially inactivates mTORC1, the influx of amino acids derived from lysosomal proteolysis subsequently restores its activity (*Yu et al., 2010*).

In addition to the dampening of translation associated with mTORC1 inhibition, activation of GCN2 by uncharged tRNAs also modulates cellular translational load via phosphorylation of the

eIF2α subunit of the translation initiation complex. This phosphorylation event reduces bulk translation initiation while simultaneously upregulating the translation of select adaptive factors such as ATF4 (*Lu et al., 2004*; *Vattem and Wek, 2004*). As a transcription factor, ATF4 directs an adaptive response program which increases non-essential amino acid synthesis as well as amino acid uptake, promoting cell survival and proliferation in amino-acid-limited conditions (*Kilberg et al., 2009*; *Ye et al., 2010*).

In contrast to mTORC1, which receives information about amino acid sufficiency predominantly from a sampling of specific free amino acids, GCN2 is thought to respond to an overall accumulation of uncharged tRNAs in amino-acid-limited conditions. However, it remains unknown whether limitations in amino acid supply affect pools of all charged tRNAs equally, or whether some tRNA isoacceptors are affected more than others. Here, we use a tRNA charging profiling method (CHARGE-seq) to survey the pools of charged tRNAs in amino-acid-deprived cells and investigate how amino acid deficit-associated patterns of tRNA charging affect protein synthesis.

## Results

### Amino acid deprivation leads to selective uncharging of glutamine-specific tRNAs

In agreement with previous reports, depriving immortalized mouse embryonic fibroblasts (MEFs) of all amino acids resulted in an initial loss of mTORC1 activity, as measured by the phosphorylation of its substrate, S6 kinase 1 (S6K1), yet a prolonged period of amino acid deprivation resulted in a near-complete reactivation of mTORC1 (*Figure 1A*). Consistent with the pivotal role of the lysosome as a supplier of amino acids in conditions when extracellular amino acids are depleted, reactivation of mTORC1 was suppressed by bafilomycin A1, an inhibitor of lysosomal H$^+$-ATPase.

In contrast to the rapid loss and subsequent recovery of mTORC1 activity following amino acid withdrawal, the onset of GCN2 activation in MEFs was gradual, with little autophosphorylation of Thr899 present at 1 hr post-amino acid withdrawal and then progressively increasing at subsequent time points (*Figure 1A*). The onset of GCN2 activation was mirrored by a gradual decline in the cellular translational output as reflected by the incorporation of O-propargyl-puromycin (OPP), a tRNA-mimetic compound, into nascent polypeptides (*Figure 1B*). Importantly, even after 6 hr without extracellular amino acids, there remained ongoing translation in amino-acid-deprived cells in comparison to cells treated with the translation elongation inhibitor cycloheximide, demonstrating that amino-acid-deprived cells can continue to engage in adaptive translation even while bulk translation continues to decline.

To test whether the observed suppression of translation might be attributed primarily to GCN2-driven phosphorylation of eIF2α, we treated amino-acid-deprived cells with a small molecule ISRIB, which counteracts the phosphorylated eIF2α-mediated inhibition of eIF2B, a guanine exchange factor for the eIF2 translation initiation complex (*Sidrauski et al., 2015*; *Tsai et al., 2018*). There was only a minimal effect of ISRIB on the suppression of translation in amino-acid-deprived cells (*Figure 1—figure supplement 1A*) – even though it efficiently blocked ATF4 accumulation after 6 hr of amino acid deprivation and elicited a further increase in eIF2α phosphorylation (*Figure 1—figure supplement 1B*). This observation suggests that GCN2-mediated eIF2α phosphorylation was not the major cause of the suppression of translation associated with amino acid withdrawal.

The observation that a 1-hr-long amino acid withdrawal triggers only a minimal amount of GCN2 phosphorylation suggests that cells are able to maintain the tRNA compartment charged during this period. Accordingly, the activation of GCN2 seen upon persistent amino acid deprivation may indicate that the cell's ability to rely on alternative amino acid sources has been exceeded, resulting in an eventual loss of tRNA charging. This raised a further question: does GCN2 activation observed upon prolonged amino acid deprivation reflect the uncharging of an entire tRNA compartment en masse or of select tRNA species only? To address these questions, we measured the charged status of various tRNA species after a short-term (1 hr) vs. a prolonged (6 hr) amino acid withdrawal. Specifically, we employed a sodium periodate oxidation method followed by the ligation of a 3′end of tRNA to a DNA adaptor (*Rizzino and Freundlich, 1975*; *Zaborske et al., 2009*). Periodate treatment destroys the ribose ring on a 3′ end of an uncharged tRNA, while a 3′ end of a charged tRNA remains protected from oxidation by an amino acid covalently bound to it (*Figure 1—figure*

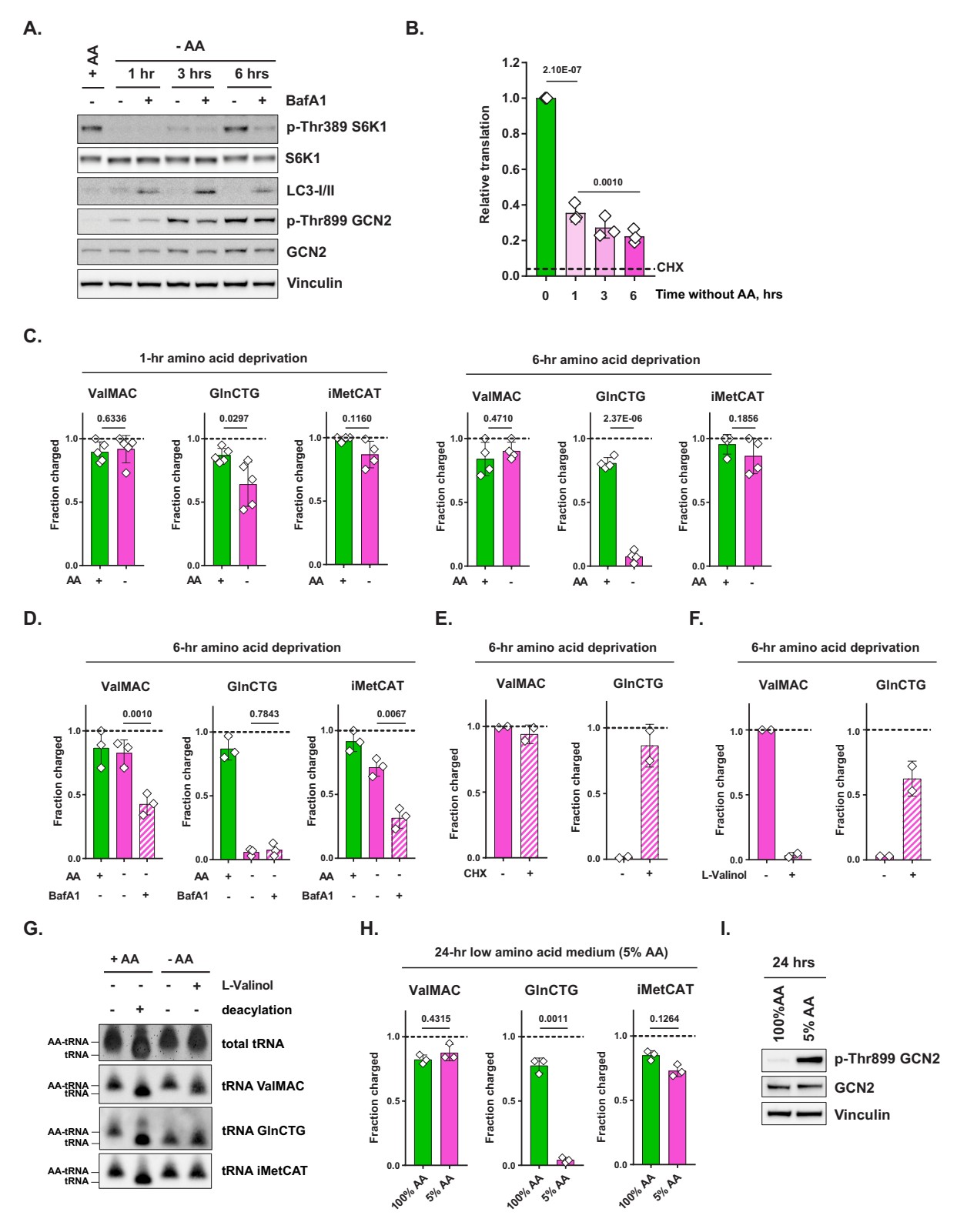

**Figure 1.** Amino acid deprivation triggers selective uncharging of tRNA^Gln. (**A**) Mouse embryonic fibroblasts (MEFs) were treated with amino-acid-free DMEM for indicated periods of time in presence (+) or absence (-) of 100 nM bafilomycin A1 (BafA1). Cell lysates were analyzed by western blotting. A representative result (out of three independent experiments) is shown. (**B**) MEFs were treated with amino-acid-free DMEM for indicated periods of time. Translational activity was determined by measuring O-propargyl-puromycin (OPP) incorporation. For CHX-treated control, cells were treated with 10
*Figure 1 continued on next page*

*Figure 1 continued*

µg/mL of cycloheximide (CHX) for 10 min prior to adding OPP. Quantification of OPP incorporated into nascent polypeptides was performed by labeling permeabilized cells with an azide-coupled fluorophore, followed by FACS analysis. Data are shown as mean ± SD of N = 3 biological replicates (defined as independent experiments that were carried out in distinct points in time). (C) MEFs were treated with amino-acid-free DMEM for 1 or 6 hr and tRNA charging of indicated isodecoder groups was measured. Primers specific to ValAAC/ValCAC (ValMAC), GlnCTG and initiator MetCAT were used. The last three letters denote an anticodon or a group of anticodons targeted. Data are shown as mean ± SD of N = 4–5 biological replicates (defined as samples that were plated, treated and assayed as part of experiments conducted independently in disparate points in time). (D) MEFs were treated with complete or amino-acid-free DMEM for 6 hr in presence (+) or absence (-) of 100 nM bafilomycin A1 (BafA1) and subjected to tRNA charging assay. Data are shown as mean ± SD of N = 3 biological replicates. (E, F) MEFs were treated with amino-acid-free DMEM for 6 hr; 10 µg/mL cycloheximide (E) or 2 mM L-valinol (F) was added for the last 10 min (for cycloheximide) or 30 min (for L-valinol) as shown. tRNA charging of indicated isodecoder groups was measured. Data are shown as mean ± SD of N = 2 biological replicates. (G) MEFs were treated with complete or amino-acid-free DMEM for 6 hr. Where indicated, 2 mM L-valinol was added for the last 30 min of treatment. tRNA samples were harvested and deacylated control was prepared by incubating in pH = 9 Tris buffer. tRNA samples were run on acetate-urea PAGE gel, transferred to Hybond N+ membranes and probed with biotin-conjugated oligo probes specific for indicated tRNA isoacceptors or groups of isoacceptors. A representative result (out of three independent experiments) is shown. (H) MEFs were treated with 100% AA (complete DMEM) or 5% AA (each amino acid supplied at a 5% of standard DMEM formulation) medium for 24 hr and subjected to tRNA charging assay. Data are shown as mean ± SD of N = 3 biological replicates. (I) MEF samples treated as in (H) were analyzed by western blot. A representative result (out of three independent experiments) is shown. p-Values were calculated by one-way ANOVA with Holm-Sidak post-test (B,D) or by paired Student's t test (C,H). See also *Figure 1—figure supplement 1*. The online version of this article includes the following source data and figure supplement(s) for figure 1:

**Source data 1.** Summary data and statistics for O-propargyl-puromycin assays, tRNA charging assays and mass spectrometry measurements presented in *Figure 1* and *Figure 1—figure supplement 1*.

**Figure supplement 1.** Additional data on amino acid sensing and tRNA charging in amino-acid-depleted cells.

*supplement 1C*). Since only those tRNAs with intact 3′ ends can be ligated to a DNA adaptor, comparing the relative abundance of tRNA-DNA hybrids in a periodate-treated vs. untreated samples via RT-qPCR provides a quantitative assessment of the proportion of charged tRNA in a sample. Using qPCR primers specific for select groups of tRNA isodecoders (i.e. groups of tRNAs that share the same anticodon), we measured the charged status of select tRNAs specific for initiator methionine, leucine, arginine, valine, and glutamine. To verify the on-target nature of tRNA-specific primers used in this study, the resulting qPCR products were subcloned into a TOPO vector and their identity was confirmed by sequencing.

The charging of amino acids on initiator tRNA$^{Met}$, tRNA$^{Leu}$, tRNA$^{Arg}$, and tRNA$^{Val}$ was maintained not only after 1 hr, but even after 6 hr of amino acid deprivation (*Figure 1C*, *Figure 1—figure supplement 1D*). In contrast, tRNA$^{Gln}$ charging was significantly reduced at 1 hr ($p < 0.05$) and was almost completely uncharged at 6 hr ($p < 0.001$) (*Figure 1C*). Mass spectrometry analysis revealed that intracellular free glutamine levels declined more than other amino acids as well (*Figure 1—figure supplement 1E*). Amino acid charging of methionine, arginine, and leucine tRNAs was maintained despite the fact that we measured significant reductions in their intracellular free amino acid levels 1 hr after cells were placed in amino-acid-deficient medium (*Figure 1—figure supplement 1E*).

We reasoned that the ability of cells to maintain the charge on most tRNAs in the absence of extracellular free amino acids might be dependent upon the use of various protein substrates as an alternative source of free amino acids and thus must require intact lysosomal function. In support of this prediction, when cells were deprived of amino acids for 6 hr in the presence of bafilomycin A1 to impair lysosomal function, the charge on tRNA$^{Val}$ and initiator tRNA$^{Met}$ pools could no longer be maintained (*Figure 1D*). Notably, bafilomycin A1 treatment did not lead to a further increase in GCN2 phosphorylation, which may indicate that the amount of uncharged tRNA$^{Gln}$ present in the cell at the 6 hr time point is sufficient to reach the upper limit of the tRNA-sensing capacity of GCN2.

Next, we asked whether the depletion of charged tRNA$^{Gln}$ in amino-acid-deprived cells is a consequence of its charging becoming inhibited in amino-acid-deprived cells via an unknown mechanism, or, alternatively, tRNA$^{Gln}$ continues to be charged with free glutamine, yet the charged form becomes consumed in protein synthesis and as a result, does not accumulate. If the latter is the case, inhibiting ongoing translation should rapidly restore tRNA$^{Gln}$ to its charged state. To test this hypothesis, we added cycloheximide, a translation elongation inhibitor, or L-valinol, a competitive inhibitor of valyl-tRNA synthetase, for, respectively, the last 10 min or 30 min of a 6 hr amino acid

deprivation treatment and assessed the effect of these inhibitors on tRNA$^{Gln}$ charging. Indeed, both cycloheximide and L-valinol promoted accumulation of the charged form of tRNA$^{Gln}$ (*Figure 1E,F*). In addition, L-valinol, but not cycloheximide, resulted in the loss of charge on tRNA$^{Val}$, which is consistent with its activity as a competitive inhibitor of valyl-tRNA synthetase (*Figure 1F*). The rapid accumulation of the charged form of tRNA$^{Gln}$ upon translation inhibition indicates that the charged form of tRNA$^{Gln}$ continues to be produced in amino-acid-deprived cells but is rapidly consumed in protein synthesis, which is consistent with the observation that translation continues to take place in cells completely deprived of extracellular free amino acids (*Figure 1B*). However, the results suggest that neither the synthesis of glutamine nor its recovery through protein degradation is sufficient to maintain tRNA charging in the absence of an extracellular supply of glutamine.

We sought to independently confirm our findings with a northern hybridization-based tRNA charging assay (*Figure 1G*; *Jester, 2011*). In this assay, the charged state of individual tRNA isodecoders (or groups of tRNA isodecoders) is determined differently from the periodate method. Specifically, the assay takes advantage of the differences in electrophoretic mobility of charged vs. uncharged tRNA molecules. In agreement with the results obtained via the periodate-based assay, we found that tRNA$^{Gln}$, but not tRNA$^{Val}$ or tRNA$^{iMet}$ became uncharged in cells that were deprived of amino acids for 6 hr, whereas addition of L-valinol has partially recovered the charged form of tRNA$^{Gln}$.

Glutamine is the most abundant extracellular amino acid. Thus, a more pronounced depletion of the charged form of tRNA$^{Gln}$ as well as intracellular free glutamine in comparison to other amino acids may simply reflect that the absolute reduction in glutamine levels is much higher than that of other amino acids when cells are acutely depleted of all extracellular amino acids. Furthermore, a short-term, complete withdrawal of extracellular amino acids is unlikely to be representative of most amino-acid-poor environments that cells might encounter in vivo. In an effort to model a more physiologically relevant scenario, we asked whether a chronic exposure to reduced quantities of amino acids would similarly lead to a selective depletion of tRNA$^{Gln}$ but not of other tRNAs. To address this issue, we cultured MEFs in a medium formulation in which each amino acid was supplied at the level of 5% of that in standard DMEM. Even under these conditions where extracellular glutamine is at 200 µM, we found that tRNA$^{Gln}$ charging was almost completely lost (p=0.0011), while other tRNA isoacceptors remained charged (*Figure 1H*). The depletion of the charged form of tRNA$^{Gln}$ under these conditions was accompanied by an increase in GCN2 phosphorylation (*Figure 1I*).

## tRNA$^{Gln}$ charging can be restored by glutaminase inhibition

Glutamine is not only incorporated into proteins but also serves as a precursor for numerous other biosynthetic pathways (*Zhang et al., 2017*), which may explain the disproportional depletion of its charged tRNA when extracellular levels of amino acids are reduced. In order to dissect the metabolic basis of this phenomenon, we first asked whether the disproportional depletion of charged tRNA$^{Gln}$ may be explained by the glutamine utilization in the biosynthesis of nucleotides and other macromolecules, and thus may be linked to the proliferative status of a cell. However, amino acid deprivation in quiescent MEFs resulted in a similar pattern of GCN2 induction (*Figure 1—figure supplement 1F*) as well as in a selective loss of charged tRNA$^{Gln}$ (*Figure 1—figure supplement 1G*), indicating that the inability of cells to maintain tRNA$^{Gln}$ charge in amino-acid-deprived state was independent of the cell's proliferative status.

Next, we tested whether the disproportional depletion of charged tRNA$^{Gln}$ may be due to the consumption of glutamine in glutaminase (GLS)-driven TCA cycle anaplerosis (*Figure 2A*). When initial results suggested that glutaminase inhibition could restore tRNA$^{Gln}$ charging, we undertook systematic profiling of the changes in the charged states across the entire cytosolic tRNA compartment. To this end, we have adapted the tRNA charging assay for a high-throughput format (CHARGE-seq), in which cDNA generated from tRNA-DNA hybrids derived from periodate-treated and control samples serves as a template for the library generation for multiplexed Illumina sequencing (*Figure 2B*). This method allowed us to reliably detect 49 out of 50 known cytosolic tRNA isodecoders and determine their charge status (a tRNA for IleGAT returned a low number of reads and was excluded from the analysis). Using this assay, we profiled the charged state of individual tRNA isodecoders following a 6 hr period of amino acid withdrawal in presence or absence of CB-839, an allosteric inhibitor of glutaminase in order to investigate whether glutaminase-driven catabolism of glutamine may contribute to the disproportionate depletion of the charged form of tRNA$^{Gln}$.

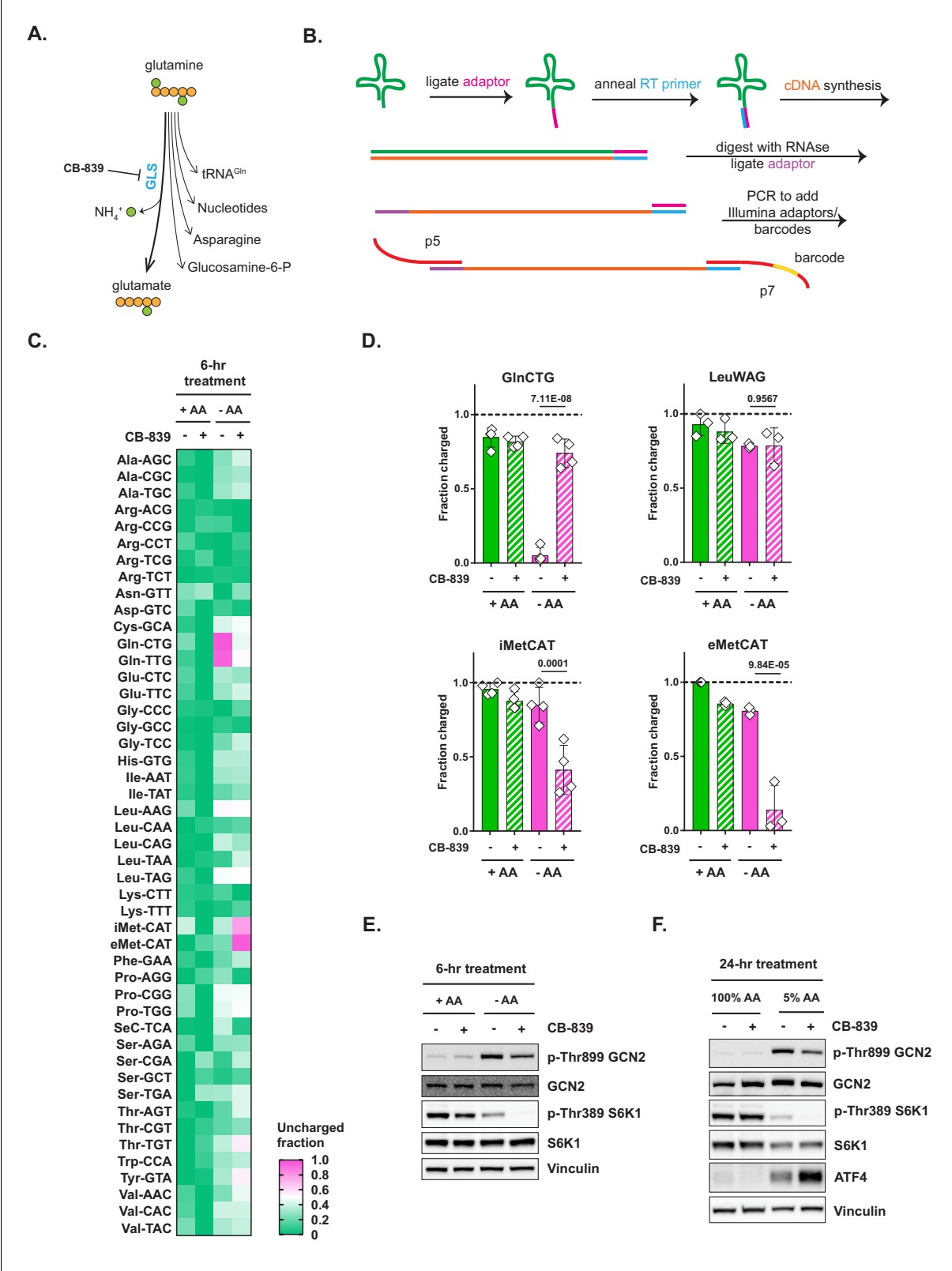

**Figure 2.** Glutaminase inhibition restores charged tRNA$^{Gln}$ pools in amino-acid-deprived cells. (A) Depiction of glutamine utilization pathways. (B) A method for a high-throughput profiling of tRNA charging (CHARGE-seq). (C) Mouse embryonic fibroblasts (MEFs) were treated with complete or amino-acid-free DMEM for 6 hr in presence (+) or absence (-) of 1 μM glutaminase inhibitor CB-839 and analyzed by CHARGE-seq. A representative result (out of two independent experiments) is shown. (D) MEFs were treated as in (C) and subjected to tRNA charging assay with qPCR as a readout. *Figure 2 continued on next page*

*Figure 2 continued*

Data are shown as mean ± SD of N = 3 biological replicates. p-Values were calculated by one-way ANOVA with Holm-Sidak post-test. (**E, F**) Western blots of lysates from MEFs treated with complete or amino-acid-free DMEM for 6 hr (**E**), or with 100% AA (complete DMEM) or 5% AA (each amino acid supplied at a 5% of standard DMEM formulation) DMEM (**F**) for 24 hr, in presence (+) or absence (-) of 1 µM glutaminase inhibitor CB-839. A representative result (out of three independent experiments) is shown. See also *Figure 2—figure supplement 1*.

The online version of this article includes the following source data and figure supplement(s) for figure 2:

**Source data 1.** Summary data and statistics for tRNA charging and cell proliferation assays presented in *Figure 2* and *Figure 2—figure supplement 1*.

**Figure supplement 1.** Loss of tRNA$^{Gln}$ charging and its restoration via the inhibition of glutaminase across a variety of cellular contexts.

The CHARGE-seq readout confirmed that tRNA$^{Gln}$ becomes selectively uncharged in amino-acid-deprived cells. In fact, the assay revealed that the two known tRNA$^{Gln}$ isodecoders – those corresponding to glutamine codons CAG and CAA – were the sole class of cytosolic tRNAs that became uncharged in amino-acid-deprived cells (*Figure 2C*). Furthermore, glutaminase inhibitor CB-839 rescued the charge on both tRNA$^{Gln}$ isodecoders in amino-acid-depleted cells, while having no discernible effect on tRNA charging in amino-acid-replete cells. Interestingly, we also found that the rescue of charged tRNA$^{Gln}$ by CB-839 was accompanied by the uncharging of initiator and elongator tRNA$^{Met}$ species, as well as a reduced charging of several other tRNAs, which might indicate that tRNA$^{Met}$ is the second-most limiting tRNA class in MEFs subjected to amino acid deprivation.

To confirm the CHARGE-seq results, we also utilized the qPCR method described above, and observed a similar rescue of tRNA$^{Gln}$ charging by CB-839 and a reciprocal uncharging of tRNA$^{Met}$ (*Figure 2D*). Consistent with the observed CB-839-triggered depletion of charged tRNA$^{Met}$, GLS inhibition had only a marginal effect on GCN2 phosphorylation in MEFs deprived of amino acids for 6 hr (*Figure 2E*), or when cells were cultured for 24 hr in a 5% amino acid DMEM formulation, in which each amino acid contained in standard DMEM was supplied at a fraction equaling 5% of that in DMEM (*Figure 2F*).

To determine if our findings were generalizable to other cell types and species, we examined tRNA charging in human cell lines and detected the same pattern of selective depletion of charged tRNA$^{Gln}$ and its rescue by CB-839. As presented in *Figure 2—figure supplement 1*, A498, a renal cell line, and MiaPaCa2, a pancreatic cell line, show selective depletion of tRNA$^{Gln}$ when deprived of extracellular amino acids that can be recovered by treatment with CB-839 (*Figure 2—figure supplement 1A,B*).

Interestingly, there was no uncharging of tRNA$^{Met}$ in A498 or MiaPaCa2 when tRNA$^{Gln}$ charging was restored in cells cultured in amino-acid-deficient medium. Instead, uncharging of tRNA$^{Arg}$ was detected in MiaPaCa2 cells (*Figure 2—figure supplement 1B*), but not in A498 cells (*Figure 2—figure supplement 1A*). These observations indicate that the limiting nature of various charged tRNA pools may be dictated by the cellular context and differences in metabolism of individual amino acids in these cells. The specific loss of charged tRNA$^{Met}$ in amino-acid-deprived MEFs upon restoring the charged form of tRNA$^{Gln}$ could be potentially explained by the critical role of methionine in S-adenosyl-methionine (SAM) production in support of cellular methylation and glutathione production through the transsulfuration pathway (*Zhu et al., 2019*).

In contrast, in MiaPaCa2, a pancreatic ductal adenocarcinoma (PDAC) cell line, the rescue of tRNA$^{Gln}$ charging triggered the uncharging of tRNA$^{Arg}$ instead. This finding may indicate that arginine is more profoundly limiting in amino-acid-depleted MiaPaCa2 cells than even methionine. This effect could be explained by the fact that MiaPaCa2, as are many other PDAC-derived lines, is deficient for argininosuccinate synthetase (ASS1), and is, as a result, an arginine auxotroph (*Singh et al., 2019*). The inability of MiaPaCa2 cells to produce arginine de novo from citrulline and aspartate may thus explain the observed phenomenon that as soon as tRNA$^{Gln}$ charging is restored in these cells, arginine becomes the most limiting substrate for tRNA charging.

The effect of the rescue of tRNA$^{Gln}$ charging by CB-839 in amino-acid-deprived A498 cells was intriguing, as it leads to neither tRNA$^{Met}$ nor tRNA$^{Arg}$ losing their charge. Of note, the observed lack of compensatory tRNA uncharging is mirrored by a near-complete reversal of amino acid depletion-triggered GCN2 activation in A498 cells (*Figure 2—figure supplement 1C,D*) in contrast to MEFs (*Figure 2E,F*) and MiaPaCa2 cells (*Figure 2—figure supplement 1E,F*). The metabolic basis of this effect remains to be investigated. Taken together, the heterogeneity of responses of the tRNA

compartment to charged tRNA$^{Gln}$ restoration in amino-acid-depleted cells may be an important new factor in understanding the amino acid requirements of different cell types.

In agreement with published studies, CB-839 inhibited proliferation of MEFs (*Figure 2—figure supplement 1G*), A498 (*Figure 2—figure supplement 1H*), and MiaPaCa2 (*Figure 2—figure supplement 1I*) cells in complete (100% AA) medium. However, the cell lines were not sensitive to glutaminase inhibition when cultured in low AA DMEM (5% AA). In fact, CB-839 treatment enabled proliferation of A498 cells (*Figure 2—figure supplement 1H*) in 5% AA DMEM. Collectively, these observations suggest that while glutaminase activity is beneficial for fueling biosynthesis and cell proliferation when amino acids are plentiful, it may, in fact, become a liability for cells that encounter amino-acid-poor environments. Indeed, we detected that cells maintain a robust expression of glutaminase regardless of the status of their amino acid supply (*Figure 2—figure supplement 1J*), further supporting the key role of glutaminase in depleting charged tRNA$^{Gln}$ in amino-acid-poor conditions.

CB-839 treatment also suppressed the eventual reactivation of mTORC1 upon a prolonged amino acid deprivation in all cell types tested (*Figure 2E,F* and *Figure 2—figure supplement 1C–F*), while no effect of CB-839 on mTORC1 activity in presence of amino acids was evident. One possible explanation of this effect is that by restoring charged tRNA$^{Gln}$ pools, inhibition of glutaminolysis may, in fact, bolster adaptive translation in amino-acid-depleted environments. This increase in translation may deplete the levels of essential amino acids such as methionine (*Figure 2D*) and arginine (*Figure 2—figure supplement 1B*), needed to activate mTORC1.

Intriguingly, CB-839 treatment was associated with an increase in ATF4 accumulation in amino-acid-poor conditions across all cellular contexts tested (*Figure 2F*, *Figure 2—figure supplement 1D*, *Figure 2—figure supplement 1F*), despite eliciting no further increase in GCN2 phosphorylation. Translational upregulation of ATF4, a master coordinator of amino acid stress response, is known to be driven by the GCN2-mediated eIF2α phosphorylation in a highly specific manner (*Dever et al., 1992*; *Vattem and Wek, 2004*). However, ATF4 protein is also a very short-lived protein with a half-life of only 1 hr (*Lassot et al., 2001*) – which, paradoxically, makes its accumulation highly dependent on the cellular ability to carry out translation. Taking this into account, the increase in ATF4 accumulation in amino-acid-depleted cells upon CB-839 treatment may indicate that restoring tRNA$^{Gln}$ charging augments the total cellular capacity for translation – which, in turn, allows cells to accumulate more ATF4. In view of this observation, we set out to further explore how restoration tRNA$^{Gln}$ charging affects translation.

## Glutaminase inhibitors augment translation in amino-acid-deprived cells

The observation that glutaminase inhibitor treatment boosted ATF4 levels in amino-acid-deprived cells has prompted us to examine the effect of glutaminase inhibitors on bulk translation in these conditions. To this end, we measured the amount of OPP incorporation in MEFs subjected to amino-acid-replete and -deficient media in presence or absence of CB-839. Indeed, glutaminase inhibition markedly increased bulk translation in amino-acid-deprived cells (*Figure 3A*). A similar effect was observed with other available GLS inhibitors – BPTES and Compound 968 (*Figure 3B*). Moreover, supplying L-glutamine at the concentration of 200 µM (which represents 5% of DMEM formulation) facilitated translation in amino-acid-deprived cells to an extent similar to that seen upon GLS inhibition (*Figure 3C*). In contrast, supplying 10 µM L-methionine (5% of DMEM formulation) had no effect on the overall translation rate, indicating that translation in amino-acid-deprived cells is hampered specifically by the deficit of charged tRNA$^{Gln}$, which must be resolved before any benefit could be derived from resupplying other limiting amino acids. Similarly to the effects observed in MEFs, either glutaminase inhibition or supplementation of exogenous glutamine rescued amino acid withdrawal-associated suppression of translation in MiaPaCa2 (*Figure 3—figure supplement 1A*) and A498 (*Figure 3—figure supplement 1B*) cells as well. This result indicates that the loss of tRNA$^{Gln}$ charging is a key limiting factor for new protein synthesis in amino-acid-poor conditions across a variety of cellular contexts.

Although the OPP incorporation assay provides a general measure of translational activity, it does not necessarily reflect a capacity of a cell to successfully produce complete, functional proteins – a property critical for maintaining cell viability, avoiding proteotoxic stress and allowing translation of stress-adaptive factors. To assess the capacity of cells to produce functional proteins in amino-acid-deprived conditions, we utilized a retrovirally encoded stable GFP reporter (half-life ~26 hr) and a destabilized GFP reporter, in which a GFP ORF was fused to a degron of mouse ornithine

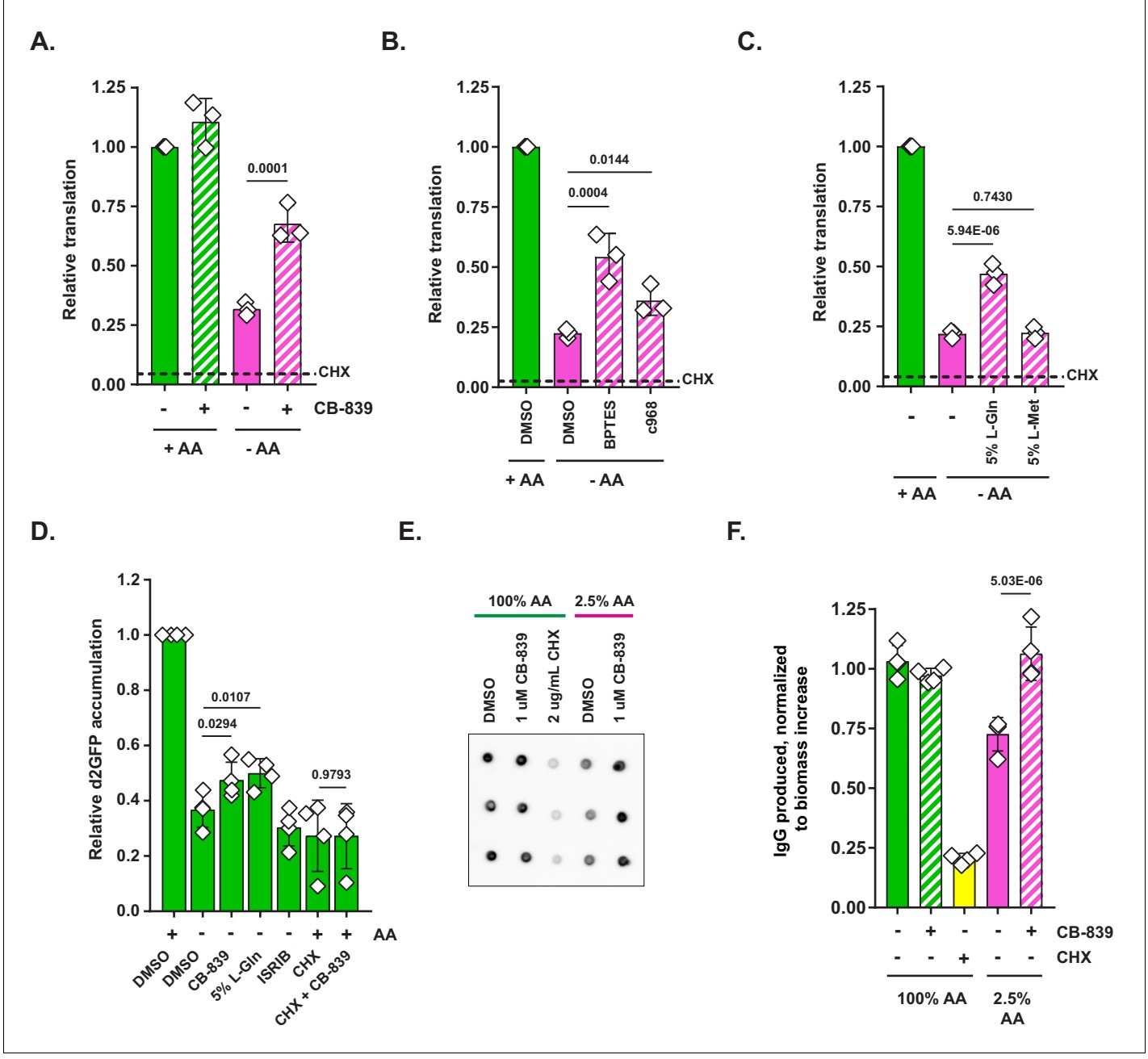

**Figure 3.** Glutaminase inhibition facilitates protein synthesis when amino acids are limiting. (**A**) Mouse embryonic fibroblasts (MEFs) were treated with complete or amino-acid-free DMEM for 6 hr in presence (+) or absence (-) of 1 μM glutaminase inhibitor CB-839. Translational activity was determined by measuring O-propargyl-puromycin (OPP) incorporation. Dotted line indicates relative amount of OPP incorporation in cells pretreated with 10 μg/mL cycloheximide (CHX). Data are shown as mean ± SD of N = 3 biological replicates. (**B**) MEFs were treated with complete or amino-acid-free DMEM in presence of 10 μM BPTES, 10 μM Compound 968 or DMSO as a control. Translation was assayed as in (**A**). Data are shown as mean ± SD of N = 3 biological replicates. (**C**) MEFs were treated with complete or amino-acid-free DMEM in presence of 200 μM L-glutamine (5% of DMEM formulation) or 10 μM L-methionine (5% of DMEM formulation). Translation was assayed as in (**A**). Data are shown as mean ± SD of N = 3 biological replicates. (**D**) MEFs transduced with an inducible d2GFP construct (a GFP ORF fused to a degron of mouse ornithine decarboxylase) were treated with complete or amino-acid-free DMEM with doxycycline for 6 hr in presence of indicated treatments. GFP fluorescence was measured by FACS. Data are shown as mean ± SD of N = 4 biological replicates. (**E**) A20 B-cell lymphoma cells were stimulated with 4 μg/mL concanavalin A for 72 hr in complete RPMI to induce immunoglobulin synthesis. Concanavalin-A-stimulated cells were treated with 100% AA RPMI or 2.5% AA RPMI (in which each amino acid was supplied at 2.5% of standard RPMI formulation) in presence or absence of 1 μM glutaminase inhibitor CB-839 or 2 μg/mL of cycloheximide (CHX) as indicated for 24 hr. Contents of wells were harvested and centrifuged to remove cells. Supernatant volumes were corrected for differences in cellular biomass accumulation over the treatment course. Biomass-corrected supernatants were applied onto a nitrocellulose paper and blotted with an anti-

*Figure 3 continued on next page*

*Figure 3 continued*

mouse HRP-conjugated antibody. Data are shown as mean ± SD of N = 4 biological replicates. (**F**) Quantification of the dot blot from (**E**). p-Values were calculated by one-way ANOVA with Holm-Sidak post-test (**A–D, E**). See also *Figure 3—figure supplement 1*.

The online version of this article includes the following source data and figure supplement(s) for figure 3:

**Source data 1.** Summary data and statistics for O-propargyl-puromycin assays, GFP reporter assays and IgG secretion assay presented in *Figure 3* and *Figure 3—figure supplement 1*.

**Figure supplement 1.** Additional data on translational capacity of amino-acid-deprived cells across a variety of cellular contexts.

decarboxylase (d2GFP, half-life ~2 hr) (described in *Li et al., 1998*). Due to its short half-life, d2GFP must be continuously synthesized in order to be maintained at a stable level within the cell. Indeed, a 6 hr amino acid deprivation led to a precipitous decline in d2GFP signal, while having no effect on the stable GFP levels (*Figure 3—figure supplement 1C*). This effect could not be explained by an increase in clearance of d2GFP from amino-acid-deprived cells, as measured by a cycloheximide chase (*Figure 3—figure supplement 1D*).

Consistent with the observed effect on OPP incorporation, adding CB-839 or 5% L-Gln to amino-acid-deprived cells resulted in an increase in d2GFP accumulation (*Figure 3D*). No such effect was observed when amino-acid-deprived cells were treated with eIF2B activator ISRIB. This suggests that the effect of CB-839 on GFP accumulation is not a consequence of the relief of the phospho-eIF2α-driven translational inhibition. Finally, CB-839 treatment did not prevent the clearance of d2GFP from cells in which translation was arrested with cycloheximide, suggesting that CB-839-driven increase in d2GFP was not due to a non-specific interference of CB-839 with the protein degradation machinery (*Figure 3D*).

To further validate the conclusion that glutaminase inhibition augments translational capacity in amino-acid-limiting conditions, we asked whether synthesis of secreted proteins by a dedicated protein-producing cell, such as a plasma cell, which is estimated to secrete its entire protein weight in immunoglobulin per day, would be similarly bolstered by glutaminase inhibitors in amino-acid-limited conditions. To this end, we stimulated a B-cell lymphoma A20 cell line with concanavalin A, a plant-derived lectin, to induce immunoglobulin production (*Kim and Dejoy, 1986*) and compared the IgG output in amino-acid-replete (100% AA RPMI) versus amino-acid-limited (2.5% AA RPMI) conditions. As expected, IgG yield by A20 cells was reduced when amino acids were limiting, yet a concomitant inhibition of glutaminase with CB-839 restored it to a level comparable to that seen in amino-acid-replete cells (*Figure 3E,F*). Taken together, these results indicate that the selective loss of charged tRNA$^{Gln}$ limits translation in amino-acid-depleted cells, which can be counteracted by adding exogenous glutamine or blocking glutamine consumption in a glutaminolysis reaction.

## Polyglutamine-tract-containing proteins are sensitive to amino acid depletion

Glutamine accounts for approximately 4.6% of the amino acid residues within human protein sequences, which makes it neither a particularly common nor a particularly rare amino acid. However, there are 68 proteins in the human proteome that contain uninterrupted tracts of 10 or more glutamine residues, known as polyglutamine, or polyQ, tracts. PolyQ tracts are thought to provide flexibility within multisubunit protein complexes (*Tóth-Petróczy et al., 2008*) as well as to be functionally involved in phase separation phenomena (*Fiumara et al., 2010*; *Vitalis et al., 2007*). In addition, expansions of polyQ tracts in select proteins underlie the pathology of a several neurodegenerative diseases including Huntington's disease and spinocerebellar ataxia (*Lieberman et al., 2019*). Since amino acid depletion triggers a profound depletion of charged tRNA$^{Gln}$ pools, we hypothesized that translation of polyQ tracts may be particularly sensitive to amino acid depletion, which would lead to a collective downregulation of polyQ-containing proteins. Indeed, culturing MiaPaCa2 cells in amino-acid-depleted conditions (5% AA) significantly reduced the levels of several polyQ proteins, including TATA-box binding protein (TBP), mediator complex subunit 12 (MED12), core binding factor α1 (CBPα1), and CREB-binding protein (CBP). In contrast, proteins lacking polyQ tracts showed little to no depletion in 5% AA, indicating that polyQ-containing proteins are particularly sensitive to amino acid deficit (*Figure 4A*). Furthermore, supplementing amino-acid-depleted medium with either CB-839 or glutamine restored the levels of all four polyQ-

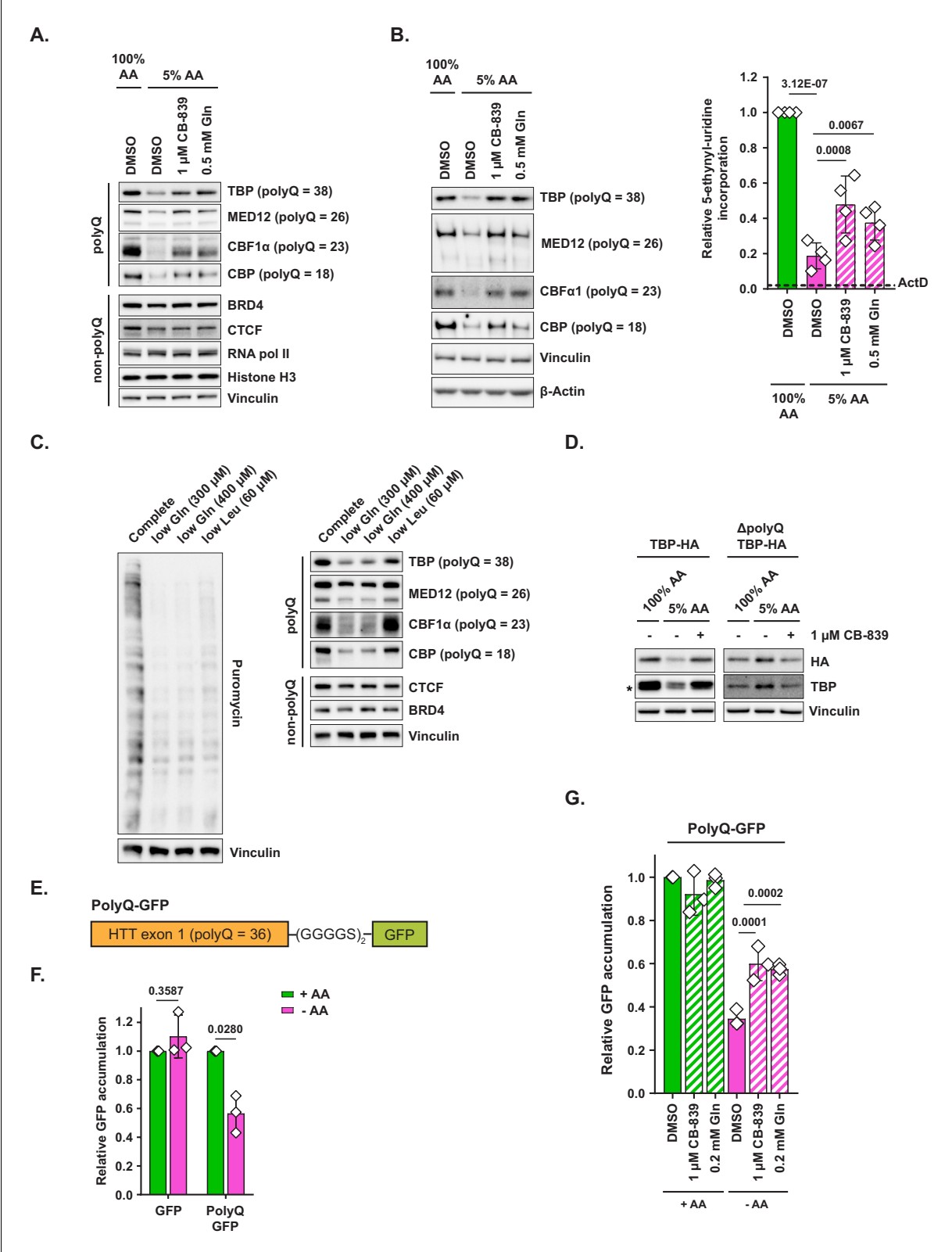

**Figure 4.** Polyglutamine-tract-containing proteins are depleted from amino-acid-deprived cells and can be recovered by glutaminase inhibition or glutamine addition. (**A**) MiaPaCa2 cells were treated as shown for 48 hr. Levels of indicated proteins were examined by western blotting. A representative result (out of three independent experiments) is shown. (**B**) MiaPaCa2 cells were treated as indicated for 72 hr and nascent RNA synthesis was monitored by 5-ethynyl-uridine (5-EU) incorporation. Levels of indicated polyQ proteins were concurrently measured by western blot. *Figure 4 continued on next page*

*Figure 4 continued*

Dotted line represents the relative value of 5-EU incorporation in cells in which transcription was arrested via actinomycin D (ActD) pretreatment for 10 min prior to adding 5-EU. Data are shown as mean ± SD of N = 4 biological replicates. (C) MiaPaCa2 cells were treated with media containing reduced quantities of glutamine or leucine or with complete medium for 48 hr, at which point new protein synthesis was assayed by puromycin incorporation. Levels of indicated polyQ and non-polyQ proteins were determined by western blotting in a parallel set of identically treated samples. A representative result (out of three independent experiments) is shown. (D) MiaPaCa2 cells were transduced with an empty retroviral vector, HA-tagged recombinant TBP (TBP-HA) or HA-tagged recombinant TBP with polyQ tract deleted (ΔpolyQ TBP-HA). Cells were treated as indicated for 48 hr, and levels of recombinant and endogenous TBP were determined by western blot. An asterisk indicates endogenous TBP. A representative result (out of three independent experiments) is shown. (E) A diagram depicting PolyQ-GFP reporter design. (F) Mouse embryonic fibroblasts (MEFs) were transduced with an inducible GFP or PolyQ-GFP construct in a retroviral vector. Cells were treated with complete or amino-acid-free DMEM in presence of doxycycline for 6 hr. GFP fluorescence was measured by FACS. Data are shown as mean ± SD of N = 3 biological replicates. (G) MEFs transduced with the PolyQ-GFP construct in a retroviral vector were treated with doxycycline-containing complete or amino-acid-free DMEM in presence of 1 µM glutaminase inhibitor CB-839, 200 µM L-glutamine + DMSO, or DMSO alone for 6 hr. GFP fluorescence was measured by FACS. Data are shown as mean ± SD of N = 3 biological replicates. p-Values were calculated by one-way ANOVA with Holm-Sidak post-test (B,G) or by paired Student's t test (F). See also *Figure 4—figure supplement 1*.

The online version of this article includes the following source data and figure supplement(s) for figure 4:

**Source data 1.** Summary data and statistics for 5-ethynyl-uridine assays and GFP reporter assays in *Figure 4* and *Figure 4—figure supplement 1*.

**Figure supplement 1.** Additional data on the recovery of expression of polyglutamine-tract-containing proteins and cellular transcriptional activity via glutaminase inhibition or glutamine supplementation.

containing proteins tested, suggesting that levels of polyQ proteins can be modulated by changes in the availability of glutamine for translation. A similar pattern was also observed in MEFs subjected to amino acid limitation (*Figure 4—figure supplement 1A,B*).

Next, we set out to determine the minimal dose of extracellular glutamine under which cells are still able to maintain the expression of polyQ proteins. Specifically, we compared the levels of polyQ proteins in cells that were exposed to either 5% AA medium (which contains 0.2 mM glutamine) or to 5% AA medium supplemented with additional glutamine, with doses ranging from 0.1 to 0.5 mM (*Figure 4—figure supplement 1C*). Indeed, glutamine concentrations found in circulation (~0.6–0.7 mM) allowed cells to maintain polyQ protein expression, those found in the peripheral regions of various solid tumors (~0.4–0.5 mM) (*Pan et al., 2016*) resulted in an intermediate level of polyQ protein expression, and finally, the concentrations found in the tumor core regions (~0.2 mM) led to the marked depletion of polyQ proteins from cells. This observation indicates that glutamine concentrations found in nutrient-poor regions within solid tumors in vivo could indeed be sufficient to deplete polyQ proteins and that the regional differences in glutamine availability may establish a pattern of heterogeneous polyQ protein expression within tumors.

PolyQ tracts are significantly more likely to be found in proteins involved in transcriptional regulation (*Albà and Guigó, 2004*) – including the four polyQ proteins tested in our study – which led us to hypothesize that collective depletion of polyQ proteins in amino-acid-limited cells may be associated with reduced RNA synthesis. Indeed, we found that exposing cells to limiting amino acid conditions markedly suppressed incorporation of labeled uridine into nascent RNA, which could be partially restored by CB-839 or added glutamine, thus mirroring the effect these conditions had on polyQ protein levels (*Figure 4B*). To further verify the on-target nature of CB-839-mediated restoration of polyQ protein expression and labeled uridine incorporation, we suppressed GLS expression in cells by RNA interference. Similarly to the effects observed with CB-839 and with exogenous glutamine supplementation, shRNA constructs targeting human GLS, but not control shRNAs, resulted in a significant rescue of both polyQ protein expression (*Figure 4—figure supplement 1D*) as well as of labeled uridine incorporation (*Figure 4—figure supplement 1E*) in amino-acid-depleted cells.

Importantly, glutaminase inhibition did not increase the size of nucleotide triphosphate pools in amino-acid-depleted cells, indicating that its effect on RNA synthesis is unlikely to be explained by increased nucleotide availability under these conditions (*Figure 4—figure supplement 1F*). Taken together, these observations raise the possibility that the presence of polyQ tracts within transcriptional regulators may represent an amino-acid-sensing adaptation that allows a cell to modulate transcriptional output in response to changes in amino acid availability.

Our observations indicate that depletion of charged tRNA$^{Gln}$ in cells cultured in amino-acid-poor conditions may compromise the cell's ability to maintain levels of polyQ-containing proteins. To test

this hypothesis further, we asked whether depletion of glutamine alone, but not of an unrelated amino acid, could trigger a decline in polyQ protein levels. To this end, we cultured cells in either glutamine-poor (300 or 400 μM L-glutamine) or in leucine-poor (60 μM L-leucine) medium for 48 hr. Even though both glutamine-poor and leucine-poor conditions equivalently inhibited bulk translation, only glutamine-poor formulations led to a significant depletion of polyQ proteins (*Figure 4C*). In a pattern similar to the one observed in 5% AA medium, non-polyQ proteins underwent little to no decline in either of these conditions. Taken together, these observations indicate that depletion of glutamine specifically, and not the inhibition of translation associated with the depletion of a different amino acid, is responsible for the observed decline in polyQ protein levels.

To further test whether polyQ tracts sensitize proteins that harbor them to amino acid deficit-associated depletion, we transduced cells with a retrovirally encoded HA-tagged TBP with its polyQ tract intact, or with a mutant form of TBP in which its 38 residue-long polyQ tract has been reduced down to two glutamine residues (ΔpolyQ TBP-HA). As expected, recombinant TBP with an intact polyQ tract behaved similarly to endogenous TBP protein – that is, its levels declined precipitously in amino-acid-depleted conditions and were fully restored by CB-839 (*Figure 4D*). In contrast, levels of recombinant TBP lacking the polyQ tract (ΔpolyQ TBP-HA) showed no decline upon amino acid depletion. Taken together, these data indicate that the polyQ tract is required for the sensitivity of TBP to amino acid depletion.

Conversely, we asked whether adding a polyQ tract to a non-polyQ protein is sufficient to make it sensitive to amino acid limitation. To this end, we designed a retrovirally encoded inducible polyQ protein reporter (PolyQ-GFP), comprised of a first exon of a mildly pathogenic allele of human huntingtin (HTT) gene (which contains a 36-residue polyQ tract) fused via a flexible linker to a GFP ORF (*Figure 4E*). Reporter expression was induced with doxycycline in presence or absence of extracellular amino acids and cells were monitored for the accumulation of GFP signal by FACS. In contrast to control GFP-expressing cells, in which the amount of fluorescence accumulation over the course of 6 hr was not affected by the absence of extracellular amino acids, PolyQ-GFP-expressing cells accumulated markedly less fluorescence in amino-acid-depleted than in complete medium (*Figure 4F*). This effect could not be attributed to the decreased half-life of PolyQ-GFP protein in amino-acid-deprived cells (*Figure 4—figure supplement 1G*). Importantly, both CB-839 and supplementing additional glutamine facilitated PolyQ-GFP accumulation in amino-acid-deprived cells, providing additional evidence that restoring charged tRNA$^{Gln}$ pools may ameliorate the defect in polyQ protein synthesis in an amino-acid-limited state (*Figure 4G*).

## Translation of polyglutamine-containing transcripts in amino-acid-depleted cells is error-prone

A recent study of translation of polyglutamine-containing human huntingtin transcript has revealed that an RNAi-mediated depletion of tRNA$^{Gln}$, or an overexpression of tRNA$^{Ala}$, leads to a decreased translational fidelity of its polyQ tract, resulting in a −1 ribosomal frameshift from a (CAG)$_n$ into a (GCA)$_n$ frame, and consequently, formation of a polyalanine-containing polypeptide product (*Girstmair et al., 2013*). Based on these findings, we hypothesized that amino acid deprivation may similarly reduce translational fidelity on polyQ tracts as a consequence of an imbalance in charged tRNA$^{Gln}$ pools relative to the rest of the tRNA compartment. To test this idea, we modified the PolyQ-GFP reporter by introducing a single base pair deletion downstream of the polyQ tract, which shifts the GFP ORF out of frame relative to the polyQ tract, and subcloned it into a lentiviral vector (*Figure 5A*). As a result of such frame mismatch, an in-frame translation of the polyQ tract does not allow GFP to be expressed, while a −1 frameshift into a (GCA)$_n$ frame repairs the GFP frame and leads to GFP accumulation.

Indeed, while PolyQ$^{-1}$-GFP-expressing MEFs cultured in complete medium expressed no GFP, amino acid deprivation triggered a marked GFP accumulation. This effect was completely blocked by CB-839 (*Figure 5B*). To test whether human cells also exhibited a loss of translational fidelity, MiaPaCa2 cells were cultured in 5% AA DMEM for 24 hr. Growth in 5% AA medium triggered GFP accumulation as well (*Figure 5C*), and this accumulation was also suppressed by CB-839 (*Figure 5C*, *Figure 5—figure supplement 1A*) or by adding exogenous glutamine (*Figure 5—figure supplement 1A*). In contrast, no frameshift-dependent translation of GFP was observed in MiaPaCa2 cells grown in complete (100% AA) medium with or without CB-839. A similar pattern was also evident in

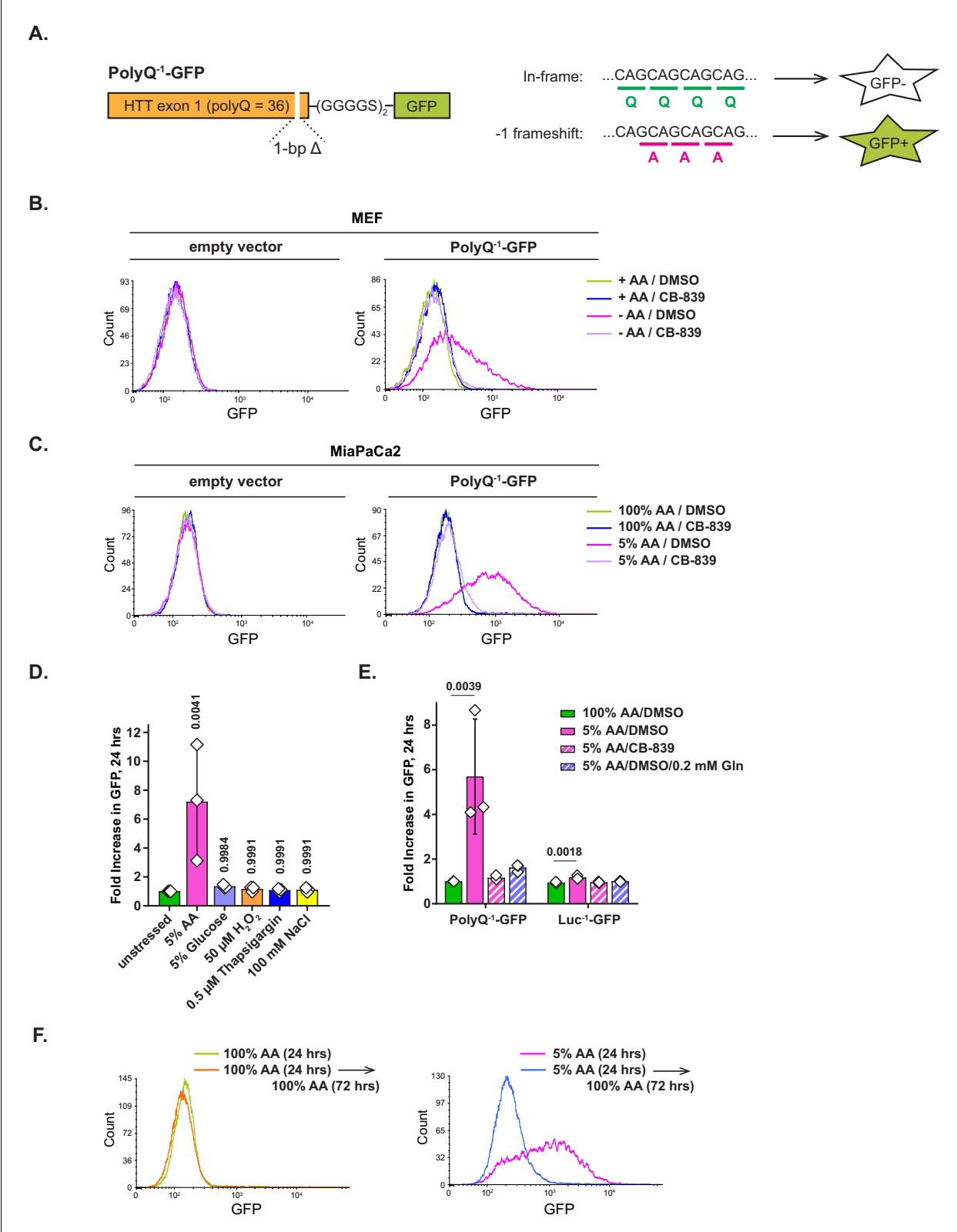

**Figure 5.** Amino acid depletion triggers frame shifting within polyglutamine-tract-containing proteins, which can be recovered by glutaminase inhibition or glutamine addition. (**A**) A diagram depicting PolyQ$^{-1}$-GFP reporter design and projected outcomes of an in-frame (relative to a $(CAG)_n$ stretch) or a −1 frame shifted translation. (**B**) Mouse embryonic fibroblasts (MEFs) transduced with PolyQ$^{-1}$-GFP reporter or empty vector control were treated with complete or amino-acid-free DMEM in presence of 1 μM glutaminase inhibitor CB-839 or DMSO for 9 hr. GFP fluorescence was measured

Figure 5 continued

by FACS. A representative result (out of three independent experiments) is shown. (C) MiaPaCa2 cells transduced with PolyQ$^{-1}$-GFP reporter or empty vector control were treated with 100% AA or 5% AA DMEM in presence or absence of 1 µM glutaminase inhibitor CB-839 for 24 hr. GFP fluorescence was measured by FACS. A representative result (out of at least three independent experiments) is shown. (D) MiaPaCa2 cells transduced with PolyQ$^{-1}$-GFP reporter were exposed to indicated stressors for 24 hr. GFP fluorescence was measured by FACS. Data are shown as mean ± SD of N = 3 biological replicates. (E) MiaPaCa2 cells transduced with PolyQ$^{-1}$-GFP or Luc$^{-1}$-GFP reporter were treated as indicated for 24 hr. GFP accumulation was measured by FACS. Data are shown as mean ± SD of N = 3 biological replicates. (F) PolyQ$^{-1}$-GFP reporter-transduced MiaPaCa2 cells were treated with 100% AA or 5% AA DMEM for 24 hr, then cultured in 100% AA medium for additional 72 hr. GFP fluorescence was measured by FACS. A representative result (out of three independent experiments) is shown. p-Values were calculated by one-way ANOVA with Holm-Sidak post-test (D, E). See also *Figure 5—figure supplement 1*.

The online version of this article includes the following source data and figure supplement(s) for figure 5:

**Source data 1.** Summary data and statistics for GFP reporter assays presented in *Figure 5* and *Figure 5—figure supplement 1*.
**Figure supplement 1.** Additional data on the amino acid depletion-triggered translational fidelity loss associated with polyglutamine tracts.

PolyQ$^{-1}$-GFP reporter-transduced murine colon carcinoma cell line MC38 (*Figure 5—figure supplement 1B*).

Notably, the extent of GFP accumulation correlated with the extent of amino acid depletion. Thus, a milder (10%) amino acid depletion also resulted in GFP accumulation within 24 hr, albeit to a lesser extent than what was observed in 5% DMEM (*Figure 5—figure supplement 1C*). In contrast to glutamine pool-restoring treatments such as glutaminase inhibition and supplementing additional exogenous glutamine, amino acid stress response inhibitor ISRIB had no effect on the GFP expression (*Figure 5—figure supplement 1D*), suggesting that the frame shifting effect is unlikely to be orchestrated by eIF2α phosphorylation-associated amino acid stress response program. Taken together, these observations indicate that across a variety of cellular contexts, amino acid depletion renders translation of polyglutamine-tract-containing transcripts prone to frame shifting. In addition, we found that inhibiting glutaminase or supplying exogenous glutamine counters the loss of translational fidelity in this reporter context, which implicates the deficit of the charged form of tRNA$^{Gln}$ in this phenomenon.

We wondered if the frame shifting effect is strictly an amino acid depletion-associated phenomenon, or if it can be triggered by other stresses as well. To this end, we subjected PolyQ$^{-1}$-GFP reporter-transduced MiaPaCa2 cells to an array of diverse cellular stresses in addition to amino acid depletion – namely, glucose depletion, oxidative stress, endoplasmic reticulum (ER) stress as well as osmotic stress (*Figure 5D*). Only amino acid depletion-associated stress resulted in GFP accumulation, indicating that amino acid deficit is a specific trigger for translational frame shifting.

To verify that the frame shifting phenotype observed in PolyQ$^{-1}$-GFP reporter-expressing cells required the polyQ tract, we created a control −1 frameshift reporter (Luc$^{-1}$-GFP), in which the HTT fragment was substituted with the N-terminal region of *Renilla* luciferase of an equal length, the sequence of which lacks polyQ tracts. In contrast to PolyQ$^{-1}$-GFP reporter-expressing cells, Luc$^{-1}$-GFP cells displayed minimal GFP accumulation in amino-acid-depleted medium (5% AA, *Figure 5E*). Furthermore, while PolyQ$^{-1}$-GFP reporter-expressing cells accumulated significantly more GFP in glutamine-poor than in leucine-poor conditions, Luc$^{-1}$-GFP cells accumulated only modest levels of GFP in either depleted medium formulation (*Figure 5—figure supplement 1E*). The effect of leucine depletion on frame shifting observed with both reporters is consistent with a nearly identical leucine content of PolyQ and Luc fragments (6 and 7 amino acid residues, respectively). Taken together, these observations indicate that the presence of a polyQ tract promotes translational frame shifting in response to amino acid depletion. Furthermore, our findings indicate that polyQ-associated frame shifting is triggered specifically by the depletion of glutamine rather than that of any given amino acid.

Finally, we asked whether the loss of translational fidelity triggered by amino acid depletion can be reversed by re-feeding the amino-acid-depleted cells with amino-acid-rich medium. Indeed, when PolyQ$^{-1}$-GFP-expressing MiaPaCa2 cells exposed to 5% AA DMEM for 24 hr were cultured in 100% AA DMEM for an additional 72 hr, GFP levels have declined markedly (*Figure 5F*), demonstrating that the frame shifting phenomenon is contingent upon amino acid deficit and is readily reversed once the adequate amino acid supply is restored.

Taken together, our observations indicate that in diverse cellular contexts, translation of polyglutamine-tract-containing transcripts is prone to frame shifting in response to amino acid deficit but not to other types of cellular stresses, and can be augmented by treatments that restore $tRNA^{Gln}$ to its charged state. These findings, in turn, led us to explore a possibility that $PolyQ^{-1}$-GFP reporter would be induced as tumor cells accumulate in vivo. To this end, we have implanted MiaPaCa2 cells transduced with $PolyQ^{-1}$-GFP reporter or empty vector control subcutaneously into nude mice. After 3 weeks, xenografts were harvested and paraffin-embedded tumor sections were stained with anti-GFP antibodies. The staining revealed multiple nests of ~10–100 $GFP^{+}$ cells residing within the

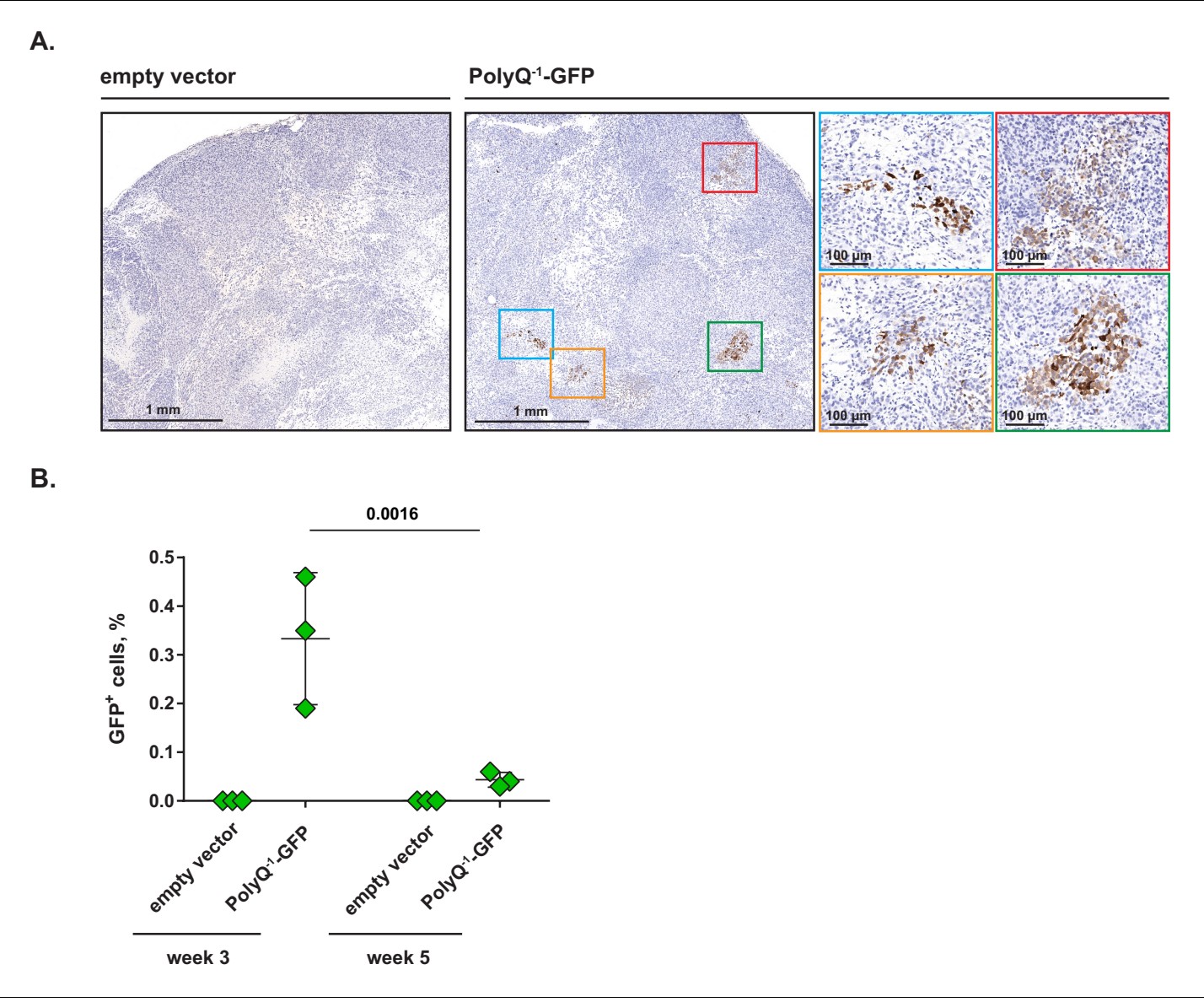

**Figure 6.** Clusters of cells undergoing frame shifting are detectable within solid tumors in vivo. (A) MiaPaCa2 cells transduced with $PolyQ^{-1}$-GFP reporter or empty vector control were injected subcutaneously into nude mice and xenografts were allowed to develop for 3 weeks. Paraffin-embedded samples were stained with anti-GFP antibody. Representative images (out of two independent experiments) are shown. (B) $PolyQ^{-1}$-GFP reporter or empty vector control-transduced MiaPaCa2 xenografts were allowed to develop for indicated periods of time, harvested and enzymatically dissociated into single-cell suspensions. Relative abundance of GFP-positive cells was determined by flow cytometry. Data are shown as mean ± SD from xenografts harvested from three animals for each time point. p-Values were calculated by two-way ANOVA with Holm-Sidak post-test (B). The online version of this article includes the following source data for figure 6:

**Source data 1.** Summary data and statistics for $GFP^{+}$ cell accumulation assay presented in *Figure 6*.

PolyQ$^{-1}$-GFP-transduced xenografts, while no staining was present in control xenografts (*Figure 6A*). To rule out the possibility that these nests might represent outgrowths of rare clones that are constitutively GFP-positive regardless of amino acid availability, we seeded PolyQ$^{-1}$-GFP – transduced MiaPaCa2 cells at a clonal density in vitro. No GFP$^+$ clones have emerged in this setting (0 out of 6 10 cm dishes assayed), indicating that groups of GFP$^+$ cells observed in vivo are unlikely to be products of rare clonal outgrowths. Taken together, our results indicate that the translational frame shifting of polyglutamine tracts can be observed in discrete areas within solid tumors in vivo.

Finally, we asked whether the abundance of GFP$^+$ cells within xenografts changes with time. To this end, we allowed PolyQ$^{-1}$-GFP or empty vector-transduced MiaPaCa2 xenografts to develop for either 3 or 5 weeks, harvested the tumors, dissociated them into single cell suspensions and determined the proportion of GFP$^+$ cells by flow cytometry. Interestingly, the proportion of GFP$^+$ cells was highest at week 3 and declined to a lower but reproducible level at week 5 (*Figure 6B*), suggesting that either GFP$^+$ cells are selected against over time or that cancer cells reduce their level of mistranslation as the xenografts mature.

## Discussion

The present work demonstrates that cells have the capacity to maintain pools of charged tRNAs during interruptions in extracellular amino acid supply. Such an ability allows the cells to retain their adaptive translational capacity for extended periods of time even when the vascular supply of free amino acids is compromised. The ability to maintain a charged tRNA compartment in amino-acid-deprived cells requires lysosomal function, which is consistent with the established role of the lysosome as a critical source of amino acids in a nutrient-limited state. Our work further reveals that adaptive translation over time results in a selective depletion of tRNA$^{Gln}$ charging that reduces the net translation.

This reduction in translation does not occur because cells cannot either recycle glutamine from lysosome-degraded proteins or take up sufficient extracellular glutamine when the vascular delivery of amino acids is compromised. Instead, it results from the propensity of cells to convert glutamine to glutamate through glutaminase. Indeed, we found that inhibiting glutaminase in the context of amino acid deficit augments the capacity of amino-acid-depleted cells to maintain tRNA$^{Gln}$ charging and sustain protein synthesis. This indicates that even though glutaminase inhibition suppresses growth when amino acids are abundant, it facilitates translation and even cell proliferation when amino acid supply is limited.

Growing cells use glutaminase to maintain TCA cycle anaplerosis (*DeBerardinis et al., 2007*), but even resting quiescent cells upregulate glutaminase during times of stress (*Hu et al., 2010*; *Suzuki et al., 2010*). This is suggested to be due to the use of glutamate for glutathione production (*Lora et al., 2004*; *Ogunlesi et al., 2004*), as an amino group donor to maintain ATF4-induced non-essential amino acid synthesis (*Harding et al., 2003*), or as a counter transport ion to maintain cystine uptake (*Sayin et al., 2017*). Thus, allosteric inhibitors of glutaminase may have contrasting effects on cells residing in abundant vs. limited amino acid environments in a wide variety of cellular contexts.

Amino acid limitation-associated depletion of tRNA$^{Gln}$ also results in a collective depletion of a number of key core transcription factors whose protein sequences contain polyglutamine tracts and consequently reduces cellular transcriptional output. Interestingly, a comparative sequence analysis across a broad spectrum of metazoan proteomes has revealed multiple instances in which the position of a polyQ tract is not tethered to a fixed position within the primary protein sequence among orthologs from diverse taxa. This observation suggests that the mere presence – rather than a specific location – of a polyQ tract may be of functional importance (*Schaefer et al., 2012*). Altogether, our observations raise the possibility that polyglutamine tracts in proteins may carry out a nutrient-sensing function by changing the protein level in response to extracellular amino acid availability. Finally, our work demonstrates that amino acid depletion is associated with the loss of translational fidelity among polyglutamine-containing transcripts, which, as we demonstrate, takes place in discrete areas of solid tumors in vivo. This, in turn, warrants further investigation of this phenomenon as a potential tool to identify and characterize amino-acid-poor cell populations from a variety of pathophysiological contexts.

# Materials and methods

**Key resources table**

| Reagent type (species) or resource | Designation | Source or reference | Identifiers | Additional information |
|---|---|---|---|---|
| Gene (*Homo sapiens*) | GLS | GenBank | Gene ID: 2744 | |
| Strain, strain background (*Mus musculus*, female) | Athymic nude mice | Envigo | Athymic Nude-Foxn1[nu], RRID:IMSR_JAX:007850 | |
| Cell line (*Mus musculus*) | Mouse embryonic fibroblasts (MEFs) | This laboratory | | SV40-immortalized; confirmed mycoplasma-free |
| Cell line (*Mus musculus*, female) | A20 | ATCC | TIB-208; RRID:CVCL_1940 | confirmed mycoplasma-free |
| Cell line (*Mus musculus*, female) | MC-38 | Dr. James Hodge laboratory | RRID:CVCL_B288 | confirmed mycoplasma-free |
| Cell line (*Homo sapiens*, male) | MiaPaCa2 | ATCC | CRL-1420; RRID:CVCL_0428 | Authenticated by STR; confirmed mycoplasma-free |
| Cell line (*Homo sapiens*, female) | A498 | Dr. James Hsieh laboratory | RRID:CVCL_1056 | Authenticated by STR; confirmed mycoplasma-free |
| Antibody | phospho-Thr899 GCN2, rabbit monoclonal | Abcam | Cat. #ab75836, RRID:AB_1310260 | (1:1000) dilution |
| Antibody | GCN2, rabbit polyclonal | Cell Signaling | Cat. #3302, RRID:AB_2277617 | (1:1000) dilution |
| Antibody | phospho-Thr-389-S6K1, rabbit monoclonal | Cell Signaling | Cat. #9234, RRID:AB_2269803 | (1:1000) dilution |
| Antibody | S6K1, rabbit monoclonal | Cell Signaling | Cat. #2708, RRID:AB_390722 | (1:1000) dilution |
| Antibody | vinculin, mouse monoclonal | Sigma-Aldrich | Cat. #V9131, RRID:AB_477629 | (1:2000) dilution |
| Antibody | ATF4, rabbit polyclonal | Santa Cruz | Cat. #sc-200, RRID:AB_2058752 | (1:250) dilution |
| Antibody | MED12, rabbit polyclonal | Bethyl | Cat. #A300-774A, RRID:AB_669756 | (1:1000) dilution |
| Antibody | TBP, rabbit polyclonal | Bethyl | Cat. #A301-229A, RRID:AB_890661 | (1:1000) dilution |
| Antibody | CBP, rabbit polyclonal | Bethyl | Cat. #A300-362A, RRID:AB_185573 | (1:1000) dilution |
| Antibody | CBFα1, rabbit monoclonal | Cell Signaling | Cat. #12556, RRID:AB_2732805 | (1:1000) dilution |
| Antibody | BRD4, rabbit monoclonal | Abcam | Cat. #ab128874, RRID:AB_11145462 | (1:1000) dilution |
| Antibody | CTCF, rabbit monoclonal | Bethyl | Cat. #A700-041-T, RRID:AB_2883994 | (1:1000) dilution |
| Antibody | RNA pol II, mouse monoclonal | Active Motif | Cat. #39497, RRID:AB_2732926 | (1:1000) dilution |
| Antibody | Histone H3, mouse monoclonal | Cell Signaling | Cat. #3638, RRID:AB_1642229 | (1:1000) dilution |
| Antibody | α-Tubulin, mouse monoclonal | Sigma-Aldrich | Cat. #T9026, RRID:AB_477593 | (1:1000) dilution |
| Antibody | β-Actin, mouse monoclonal | Sigma-Aldrich | Cat. #A5441, RRID:AB_476744 | (1:2000) dilution |
| Antibody | HA tag, mouse monoclonal | Cell Signaling | Cat. #2367, RRID:AB_10691311 | (1:1000) dilution |
| Antibody | Puromycin, mouse monoclonal | EMD Millipore | Cat. #MABE343, RRID:AB_2566826 | (1:500) dilution |

*Continued on next page*

*Continued*

| Reagent type (species) or resource | Designation | Source or reference | Identifiers | Additional information |
|---|---|---|---|---|
| Antibody | GLS, rabbit monoclonal | Abcam | Cat. #ab156876, RRID:AB_2721038 | (1:1000) dilution |
| Recombinant DNA reagent | pLKO.1-shCtrl1 | Gene Editing and Screening Core, MSKCC | SHC002 | Lentivirus-encoded non-targeting control shRNA |
| Recombinant DNA reagent | pLKO.1-shCtrl2 | Gene Editing and Screening Core, MSKCC | SHC007 | Lentivirus-encoded shRNA targeting luciferase |
| Recombinant DNA reagent | pLKO.1-shGLS-1 | Gene Editing and Screening Core, MSKCC | TRCN0000051136 | Lentivirus-encoded shRNA targeting human GLS |
| Recombinant DNA reagent | pLKO.1-shGLS-2 | Gene Editing and Screening Core, MSKCC | TRCN0000051135 | Lentivirus-encoded shRNA targeting human GLS |
| Recombinant DNA reagent | pTURN-hygro-GFP | This laboratory | | Retrovirus-encoded, dox-inducible vector expressing GFP |
| Recombinant DNA reagent | pTURN-hygro-d2GFP | This laboratory | | Retrovirus-encoded, dox-inducible vector expressing d2GFP (GFP fused to a degron of mouse ODC) |
| Recombinant DNA reagent | pTURN-hygro-PolyQ-GFP | This laboratory | | Retrovirus-encoded, dox-inducible vector expressing PolyQ-GFP (GFP) fused to a first exon of human HTT; design details in 'Reporter Design and Virus Production' under Materials and methods |
| Recombinant DNA reagent | pCDH-puro-PolyQ$^{-1}$-GFP | This laboratory | | Lentivirus-encoded frameshift reporter; design details in 'Reporter Design and Virus Production' under Materials and methods |
| Recombinant DNA reagent | pCDH-puro-Luc$^{-1}$-GFP | This laboratory | | Lentivirus-encoded frameshift reporter control; design details in 'Reporter Design and Virus Production' under Materials and methods |
| Sequence-based reagent | tRNA assay 5'-adenylated DNA adaptor | IDT DNA | | 5'-/5rApp/TGGAATTCTCGGG TGCCAAGG/3ddC /- 3' |
| Sequence-based reagent | 5'-phosphorylated DNA adaptor for CHARGE-seq | IDT DNA | | 5'-/5phos/AGATCGGAAGAGC GTCGTGTAGGGA/3ddC /- 3' |
| Commercial assay or kit | Click-iT Plus Alexa Fluor 647 Picolyl Azide Toolkit | Thermo Scientific | C10643 | For O-propargyl-puromycin and 5-ethynyl-uridine incorporation assay |
| Chemical compound, drug | L-Valinol | Sigma-Aldrich | 186708 | Used at 2 mM |
| Chemical compound, drug | Cycloheximide | Sigma-Aldrich | C4859 | Used at 10 µg/mL |
| Chemical compound, drug | CB-839 | Selleck Chemicals | S7655 | Used at 1 µM |
| Chemical compound, drug | Bafilomycin A1 | Cayman Chemical | 88899-55-2 | Used at 100 nM |
| Chemical compound, drug | BPTES | Cayman Chemical | 19284 | Used at 10 µM |
| Chemical compound, drug | Compound 968 | Cayman Chemical | 17199 | Used at 10 µM |
| Chemical compound, drug | ISRIB | Sigma-Aldrich | SML0843 | Used at 400 nM |

*Continued on next page*

*Continued*

| Reagent type (species) or resource | Designation | Source or reference | Identifiers | Additional information |
|---|---|---|---|---|
| Chemical compound, drug | O-propargyl-puromycin | Thermo Scientific | C10459 | Used at 20 µM |
| Chemical compound, drug | 5-ethynyl-uridine | Abcam | ab146642 | Used at 200 µM |
| Other | GtRNA database | PMID:26673694 | RRID:SCR_006939 | Genomic tRNA Database, http://gtrnadb.ucsc.edu |

## Cell culture

MEFs were isolated and immortalized with SV40 antigen as previously described (*Wei et al., 2001*). A498 human clear cell renal cell carcinoma cell line (RRID:CVCL_1056) was kindly provided by Dr. James Hsieh. MiaPaCa2 human pancreatic adenocarcinoma cell line (RRID:CVCL_0428) and A20 mouse B-cell lymphoma cell line (RRID:CVCL_1940) were from ATCC. MC38 mouse colon adenocarcinoma cell line (RRID:CVCL_B288) was kindly provided by Dr. James Hodge. All cell lines were verified to be mycoplasma-negative by MycoAlert Mycoplasma Detection Kit (LT07-318, Lonza). Human cell lines were authenticated via STR repeat mapping by the Integrated Genomics Operation core facility at MSKCC.

Cells were cultured at 37°C in a 5% $CO_2$ incubator. Tissue culture media were prepared by the Media Preparation Facility at MSKCC. MEFs, A498, MiaPaCa2, and MC38 cells were cultured in high-glucose DMEM supplemented with 10% FBS, and A20 cells were cultured in RPMI-1640 supplemented with 10% FBS and 50 µM β-mercaptoethanol. To induce quiescence, cells were cultured for 3 days post-confluence, with culture medium changed daily. For cell proliferation experiments, cell numbers at the start and the end of the experiment were counted in triplicates using the Multisizer 3 Coulter Counter (Beckman).

## Amino acid deprivation experiments

DMEM and RPMI-1640 lacking all or indicated amino acids were prepared by the Media Preparation Facility at MSKCC. For amino acid deprivation experiments in adherent cells, cells were rinsed with PBS, and treatment media lacking amino acids or with all amino acids present at an indicated fraction of that in a standard formulation, was added. Where indicated, 2 mM L-Valinol (186708, Sigma-Aldrich) or 10 µg/mL cycloheximide (C4859, Sigma-Aldrich) was added for the last 30 or 10 min of treatment, respectively. All treatment media were supplemented with 10% dialyzed FBS (Gemini Bioproducts, 100–108).

## Chemical inhibitors

CB-839 (S7655) was from Selleck Chemicals, bafilomycin A1 (88899-55-2), BPTES (19284) and compound 968 (17199) were from Cayman Chemical, and cycloheximide (C4859) and ISRIB (SML0843) were from Sigma-Aldrich. All chemical inhibitors were resuspended in DMSO. An equivalent amount of DMSO was added to control samples to control for any solvent-based effects.

## Western blotting

Protein extracts were prepared by using 1 × RIPA buffer (20–188, Millipore) supplemented with protease (1860932, ThermoFisher) and phosphatase inhibitors (78428, ThermoFisher). For analysis of transcriptional regulators, lysis buffer was supplemented with 1% SDS and benzonase (70746–4, EMD Millipore). Equal amounts of total protein were separated on NuPAGE Bis-Tris or Tris-Acetate (for large proteins) gels (Life Technologies), transferred to nitrocellulose membranes and subjected to Western blotting with indicated primary antibodies. The following primary antibodies were used: phospho-Thr899 GCN2 (ab75836, Abcam), total GCN2 (3302, Cell Signaling), phospho-Thr-389-S6K1 (9234, Cell Signaling), S6K1 (2708, Cell Signaling), vinculin (V9131, Sigma-Aldrich), ATF4 (sc-200, Santa Cruz), MED12 (A300-774A, Bethyl), TBP (A301-229A, Bethyl), CBP (A300-362A, Bethyl), CBFα1 (12556, Cell Signaling), BRD4 (ab128874, Abcam), CTCF (A700-041-T, Bethyl), RNA pol II (39497, Active Motif), Histone H3 (3638, Cell Signaling), α-Tubulin (T9026, Sigma-Aldrich), β-Actin

(A5441, Sigma-Aldrich), HA tag (2367, Cell Signaling), puromycin (MABE343, EMD Millipore), and GLS (ab156876, Abcam).

## O-propargyl-puromycin assay

Cells were treated as indicated. For the last 30 min of treatment, 20 µM O-propargyl-puromycin (OPP, C10459, Thermo Scientific) was added to wells. For cycloheximide control samples, 10 µg/mL cycloheximide was added to wells 10 min prior to the addition of OPP. Cells were harvested by trypsinization and fixed with methanol at −20℃, followed by permeabilization with 0.5% Triton-X in PBS. Fixed and permeabilized cells were stained using Click-iT Plus Alexa Fluor 647 Picolyl Azide Toolkit from Thermo Scientific (C10643) according to the manufacturer's instructions and analyzed by flow cytometry.

## tRNA charging assay

Cells were treated as indicated, placed on ice, rinsed once with cold PBS and lysed with cold TRIzol (15596018, Life Technologies) on ice. Lysates were shaken with chloroform 5:1, centrifuged at 18,600 g and precipitated with 2.7x volumes of cold ethanol in presence of 30 µg of GlycoBlue Coprecipitant (AM9515, ThermoFisher) overnight. Samples were resuspended in 0.3M acetate buffer (pH = 4.5) with 10 mM EDTA and precipitated again. Next day, samples were resuspended in 10 mM acetate buffer with 1 mM EDTA. Of each RNA sample, 2 µg was treated with 10 mM of either sodium periodate ('oxidized sample') or sodium chloride ('non-oxidized sample') and incubated for 20 min at room temperature in the dark. Sodium periodate was from Sigma-Aldrich (311448). Reactions were quenched with glucose for 15 min. Yeast tRNA$^{Phe}$ (R4018, Sigma-Aldrich) was added to each sample, after which samples were precipitated with ethanol. Samples were resuspended in 50 mM Tris buffer (pH = 9) and incubated for 50 min at 37℃, quenched with acetate buffer and precipitated. Finally, samples were resuspended in RNAse-free water and subjected to a ligation to a 5′-adenylated DNA adaptor (5′-/5rApp/TGGAATTCTCGGGTGCCAAGG/3ddC /- 3′), using truncated KQ mutant T4 RNA ligase 2 (M0373, New England Biolabs), for 3 hr at room temperature, according to *Loayza-Puch et al., 2016*. Reverse transcription was performed with SuperScript IV reverse transcriptase (18090050, Thermo Scientific) according to the manufacturer's instructions, with a primer complementary to the DNA adaptor. cDNA samples were subjected to qPCR with tRNA isodecoder-specific primers designed so that the forward (FW) primer was complementary to the 5′ end of the tRNA, and the reverse (RV) primer spanned the junction between the 3′ end of the tRNA and the ligated adaptor. The following primer pairs were used: ValMAC (FW: 5′-GTTTCCGTAGTGTAG TGGTTATCACGTTCG-3′, RV: 5′-GAGAATTCCATGGTGTTTCCGCCC-3′), iMetCAT (FW: 5′-AGCA-GAGTGGCGCAGCG-3′. RV: 5′-GAGAATTCCATGGTAGCAGAGGATGGTTTCG-3′), eMetCAT (FW: 5′-GCCTCSTTAGCGCAGTAGGTAG-3′, RV: 5′-GAGAATTCCATGGTGCCCCSTS-3′) GlnCTG (FW: 5′-GGTTCCATGGTGTAATGGTNAGCACTCTG-3′, RV: 5′-GAGAATTCCATGGAGGTTCCACCGAGA TTTG-3′), LeuWAG (FW: 5′-GGTAGYGTGGCCGAGCG-3′, RV: 5′-GAGAATTCCATGGCAGYGG TGGG-3′), ArgACG (FW: 5′-GGGCCAGTGGCGCAATG-3′, RV: 5′-GAGAATTCCATGGCGAGC-CAGC-3′), yPhe (FW: 5′-GCGGAYTTAGCTCAGTTGGGAGAG-3′, RV: 5′-GAGAATTCCATGG TGCGAAYTCTGTGG-3′). Primers were designed using reference tRNA sequences from GtRNA database (http://gtrnadb.ucsc.edu/, RRID:SCR_006939) (*Chan and Lowe, 2016*). Ct values obtained with primers specific for yeast tRNA$^{Phe}$ primers were subtracted from Ct values obtained with primers specific for an isodecoder of interest. The charged fraction value was calculated from a relative difference between a delta-Ct value from a non-oxidized (representing total) and oxidized (representing charged) samples for each primer pair. The qPCR amplicons were TA-cloned (TOPO TA cloning kit, 450641, Life Technologies) and sequenced to confirm the on-target nature of the primers. See also *Supplementary file 1* for a full list of oligonucleotides used.

## CHARGE-seq

For the high-throughput tRNA charging analysis, RNA samples were processed as above. After the reverse transcription step, 3′ end of cDNA was ligated to a 5′-phosphorylated DNA adaptor (5′-/5phos/AGATCGGAAGAGCGTCGTGTAGGGA/3ddC/- 3′) using T4 RNA ligase 1 (M0437, New England Biolabs) in presence of 20% PEG, 1 mM ATP and 30 mM hexammine cobalt(III) chloride (H7891, Sigma-Aldrich). Ligation products were subjected to eight cycles of PCR with primers containing 8-

mer barcodes and p5 and p7 Illumina adaptors. PCR products were run out on an agarose gel and ~190 bp bands were excised, purified and subjected to MiSeq paired-end Illumina sequencing. Unique mouse tRNA gene sequences were compiled from GtRNA database (http://gtrnadb.ucsc.edu/, RRID:SCR_006939) (*Chan and Lowe, 2016*). Sequences were appended with a 3′ CCA and used to create a reference assembly with the 'bowtie2-build' function. Paired-end 75 bp reads were trimmed of Illumina adapters and aligned to the reference assembly with 'bowtie2' run in 'very-sensitive' mode, allowing one mismatch to account for degeneracy that occurs at methylated adenine bases. Aligned reads were sorted and indexed with 'samtools' and a custom R script was used to assign gene alignments to each tRNA isodecoder. Reads were normalized by library size and the yeast phenylalanine tRNA spike-in counts prior to determining charge ratios. Raw and processed sequencing data was submitted to the Gene Expression Omnibus (GSE157276).

## Northern tRNA charging assay

The protocol was adapted from *Jester, 2011* with some modifications. Cells were treated as described before. RNA was harvested using Trizol/chloroform method and precipitated. Samples were resuspended in 0.3M acetate buffer (pH = 4.5) with 10 mM EDTA, except for 'deacylated control' sample, which was chemically deacylated by resuspending and incubating in Tris buffer (pH = 9.0) buffer, then quenched with acetate buffer. All samples were reprecipitated and resuspended in 10 mM acetate buffer with 1 mM EDTA and loaded on 6.5% acetate-urea PAGE gel (pH = 5.0). Sample loading was normalized to cell biomass. Gel was run at 4°C for 6 hr at a constant voltage (12 V/cm), after which gel was stained with SybrGOLD to visualize total tRNA, washed and transferred onto Hybond N$^+$ membranes (GE Amersham) at 40V for 2 hr at 4°C. Membranes were cross-linked with 1200 μJoules of UV and baked in a 50°C oven for 20 min. Membranes were blocked overnight in Church-Gilbert buffer (7% SDS in Na$_2$HPO$_4$ buffer (pH = 7.0), supplemented with 10 μg/mL salmon sperm DNA) at 42°C, then hybridized for 24 hr with 100 μM of 5′-biotin-labeled DNA oligo probe complementary to the 5′ end of tRNA isoacceptors (or groups of tRNA isoacceptors) of interest in Church-Gilbert buffer at 42°C. Membranes were washed with 0.1% SDS/1x SSC buffer and incubated with 0.16 μg/mL streptavidin-HRP (N100, Thermo Fisher) in Church-Gilbert buffer at room temperature for 30 min, washed and imaged in BioRad imager. Membranes were stripped with boiling 0.1% SDS for 15 min, reblocked and reprobed as needed. The following biotin-labeled probes were used: tRNA$^{GlnCTG}$: 5′-/5Biosg/CTAACCATTACACCATGGAAC-3′, tRNA$^{ValMAC}$: 5′-/5Biosg/GATAACCACTACACTACGGAA-3′, tRNA$^{iMetCAT}$: 5′-/5Biosg/GCTTCCGCTGCGCCACTCTGC-3′.

## Reporter design and virus production

GFP, d2GFP, and PolyQ-GFP reporter constructs were built in a pTURN-hygro doxycycline-inducible retroviral vector (a gift from Dr. Scott Lowe). The vector was digested with XhoI-EcoRI and gel-purified. ATG codon-less GFP and d2GFP ORFs preceded by a flexible linker (GGGGSGGGGS) and containing a 20 bp overlap with pTURN-hygro downstream of the GFP/d2GFP ORF were synthesized by Genewiz. PolyQ$^{-1}$-GFP reporter construct was built in a pCDH-CMV-MSC-EF1α-puro constitutive lentiviral vector (CD510B-1, System Biosciences). For the construction of PolyQ-GFP and PolyQ$^{-1}$-GFP reporter, the first exon of the human HTT allele (polyQ length = 36) was PCR-amplified from pBacMam2-DiEx-LIC-C-flag_huntingtin_full-length_Q36 construct (Addgene #111745, a gift from Dr. Cheryl Arrowsmith [*Harding et al., 2019*]). Primers for the PCR amplification were designed to contain a 20 bp overlap with the corresponding vector backbone (forward primer) and with the ATG-less GFP ORF preceded by a flexible linker (reverse primer). For PolyQ$^{-1}$ variant, the reverse primer was designed to create a single base pair deletion downstream of the poly-CAG tract. The Luc$^{-1}$ fragment for construction of Luc$^{-1}$-GFP reporter was synthesized by IDT DNA using AAG54094.1 *R. muelleri* luciferase sequence (base pairs 1–287), and its nucleotide sequence was optimized to remove STOP codons from the −1 reading frame. Resulting fragments were assembled into pTURN-hygro or pCDH-puro via the Gibson method (E2611, New England Biolabs). For GFP and d2GFP construct design, constructs synthesized by Genewiz were PCR-amplified with primers designed to add the ATG codon and 20 bp overlap with pTURN-hygro and omitting the flexible linker. Resulting PCR products were subcloned into pTURN-hygro via Gibson assembly. Retroviral particles were produced by cotransfecting the viral backbone plasmid together with packaging plasmids into 293T cells with polyethylenimine (23966–1, Polysciences). MEFs, MiaPaCa2 and MC38 cells were infected

with viral supernatant in presence of 8 µg/mL of polybrene (107689, Sigma-Aldrich) overnight and subjected to selection with 200 µg/mL hygromycin B (400052, EMD Millipore) for pTURN-hygro, or 2 µg/mL puromycin (P9620, Sigma-Aldrich) for pCDH-puro.

## IgG secretion assay

IgG production in the A20 suspension cell line was induced with 4 µg/mL Concanavalin A (C5275, Sigma-Aldrich) for 72 hr, after which cells were collected, gently centrifuged and plated in triplicates into indicated treatment media in 12-well dishes ($1 \times 10^6$ cells/well). Starting biomass was recorded by determining starting cell volume via Coulter Counter and multiplying it by starting cell number. Cells were incubated for 24 hr, after which contents of wells were collected and split into two halves. One half was used for counting cells and recording biomass as above. The AUC (area under curve) measure of the biomass increase was calculated using a formula from *Jain et al., 2012*. The other half of the sample was spun down to remove the cells. Cleared supernatants were adjusted with PBS to account for changes in biomass over the 24 hr treatment period, after which supernatants were applied onto a nitrocellulose membrane and probed with an HRP-conjugated anti-mouse antibody. Relative signal intensity was quantified using ImageLab software (BioRad).

## Uridine incorporation assay

Cells were treated as indicated. For the last 30 min of treatment, 200 µM 5-ethynyl-uridine (5-EU, ab146642, Abcam) was added to wells. For actinomycin D control samples, 4 µg/mL actinomycin D (A1410, Sigma-Aldrich) was added to wells 10 min prior to the addition of 5-EU. Cells were harvested by trypsinization and subjected to fixation/permeabilization in 125 mM PIPES (pH = 6.8), 10 mM EGTA, 1 mM $MgCl_2$, 3.7% formaldehyde and 0.2% Triton-X for 30 min at room temperature. Fixed and permeabilized cells were then stained using Click-iT Plus Alexa Fluor 647 Picolyl Azide Toolkit from Thermo Scientific (C10643) according to the manufacturer's instructions and analyzed by flow cytometry.

## RNA interference

pLKO.1-encoded shRNAs targeting human GLS (both GAC and KGA isoforms) as well as control shRNAs were provided by Gene Editing and Screening Core at MSKCC. Constructs were packaged into lentiviral particles and transduced into MiaPaCa2 as described earlier. The following shRNA were used: shCtrl-1 (SHC002, non-targeting control), shCtrl-2 (SHC007, luciferase-targeting control), shGLS-1 (TRCN0000051136) and shGLS-2 (TRCN0000051135).

## LC-MS/MS analysis

Cells in six-well dishes were treated as indicated, placed on ice and washed once with cold PBS, after which 1 mL/well of 80% methanol was added. Plates were kept in −80˚C for 24 hr, after which samples were scraped into eppendorf tubes and cleared by centrifugation at 14,000 rpm for 20 min at 4˚C. Cleared samples were transferred into fresh eppendorf tubes, dried for 5 hr in GeneVac evaporator and stored in −80˚C until further processing. The samples were reconstituted in 40 µL of 97:3 water:methanol containing 10 mM tributylamine and 15 mM acetic acid (mobile phase A) and incubated on ice for 20 min, vortexing every 5 min to ensure adequate re-suspension. All samples underwent one final centrifugation step (20,000x*g* for 20 min at 4˚C) to remove any residual particulate.

The reconstituted samples were subjected to MS/MS acquisition using an Agilent 1290 Infinity LC system equipped with a quaternary pump, multisampler, and thermostatted column compartment coupled to an Agilent 6470 series triple quadrupole system using a dual Agilent Jet Stream source for sample introduction. Data were acquired in dynamic MRM mode using electrospray ionization (ESI) in negative ion mode. The capillary voltage was 2000 V, nebulizer gas pressure of 45 Psi, drying gas temperature of 150˚C and drying gas flow rate of 13 L/min. A volume of 5 µL of sample was injected on to an Agilent Zorbax RRHD Extend-C18 (1.8 µm, 2.1 × 150 mm) column operating at 35˚C. The 24-min chromatographic gradient was performed using 10 mM tributylamine and 15 mM acetic acid in 97:3 water:methanol (mobile phase A) and 10 mM tributylamine and 15 mM acetic acid in methanol (mobile phase B), at a 0.25 mL/min flow rate. At the end of the 24 min, the gradient included a backflush of the analytical column for 6 min with 99% acetonitrile at a 0.8 mL/min flow

rate, followed by a 5 min re-equilibration step at 100% A (MassHunter Metabolomics dMRM Database and Method, Agilent Technologies).

Each batch was composed of four replicate samples for each group, two method blanks and one pooled sample. The pooled sample was prepared by mixing 5 µL of each sample in the batch. Three replicates of the pooled sample were injected at the start of the batch to condition the system, followed by samples in randomized order. The pooled sample was injected every 5–10 samples throughout the sequence serving as quality control for changes during the run. Analysis of the pooled samples was used to monitor the reproducibility of the system and the stability of the run over time. Data analysis was performed using Agilent MassHunter Quantitative Analysis (v. B.09.00).

## Xenograft experiments

All animal experiments were conducted under the guidance of MSKCC Institutional Animal Care and Use Committee. $5 \times 10^6$ MiaPaCa2 cells transduced with pCDH-puro lentivirus expressing PolyQ$^{-1}$-GFP reporter or empty vector control were mixed with Matrigel (BD Biosciences, 356273) at 1:1 ratio by volume and injected subcutaneously into opposing flanks of the 6- to 8-week-old female athymic nude mice (Envigo). At indicated time points, animals were sacrificed by $CO_2$ inhalation. Xenografts were collected and cut into two halves. One half was subjected to digestion with 20 µg/mL DNAse I and 200 U/mL Collagenase IV in serum-free DMEM for 1 hr at 37C, stained with propidium iodide (BD Biosciences, 556463) to exclude dead cells and analyzed by FACS. The second half was fixed overnight in 3.7% formaldehyde, paraffin-embedded, mounted on slides and stained with anti-GFP antibody (Histowiz).

## Acknowledgements

We thank the members of the Thompson laboratory for helpful discussions, the members of the Integrative Genomics Operation (MSKCC), especially Drs. Agnes Viale, Neeman Mohibullah and Cassidy Cobbs for advice and technical help with the CHARGE-seq protocol and Drs. Elisa de Stanchina and Xiaoping Chen of the Antitumor Assessment Core (MSKCC) for help with xenograft experiments.

## Additional information

### Competing interests

Craig B Thompson: is a founder of Agios Pharmaceuticals and a member of its scientific advisory board. He is also a former member of the Board of Directors and stockholder of Merck and Charles River Laboratories. He is a named inventor on patents related to cellular metabolism (see https://tinyurl.com/y35qvajq). The other authors declare that no competing interests exist.

### Funding

| Funder | Grant reference number | Author |
| --- | --- | --- |
| National Cancer Institute | P30 CA 008748 | Craig B Thompson |
| Damon Runyon Cancer Research Foundation | DRG 2234-15 | Bryan King |

The funders had no role in study design, data collection and interpretation, or the decision to submit the work for publication.

### Author contributions

Natalya N Pavlova, Conceptualization, Data curation, Formal analysis, Validation, Investigation, Visualization, Methodology, Writing - original draft, Writing - review and editing; Bryan King, Conceptualization, Data curation, Formal analysis, Funding acquisition, Investigation, Methodology, Writing - review and editing; Rachel H Josselsohn, Victoria L Macera, Santosha A Vardhana, Investigation; Sara Violante, Formal analysis, Investigation, Methodology; Justin R Cross, Supervision, Methodology; Craig B Thompson, Conceptualization, Supervision, Funding acquisition, Writing - review and editing

## Author ORCIDs

Natalya N Pavlova (ID) https://orcid.org/0000-0002-5120-4404
Rachel H Josselsohn (ID) https://orcid.org/0000-0002-6123-7744
Craig B Thompson (ID) https://orcid.org/0000-0003-3580-2751

## Ethics

Animal experimentation: All animal experiments were conducted in accordance with policies and practices approved by the Memorial Sloan Kettering Cancer Center Institutional Animal Care and Use Committee (IACUC), and were carried out following the NIH guidelines for animal welfare (animal protocol #11-03-007). Every effort was made to minimize suffering.

## Decision letter and Author response

Decision letter https://doi.org/10.7554/eLife.62307.sa1
Author response https://doi.org/10.7554/eLife.62307.sa2

# Additional files

## Supplementary files

• Supplementary file 1. Additional oligonucleotide sequences (not listed in the key resources table). Sequences of oligonucleotides used for tRNA charging assay (*Figures 1* and *2*) and for Northern blotting (*Figure 1G*) are listed.

• Transparent reporting form

## Data availability

High-throughput sequencing data have been deposited in GEO (accession code GSE157276).

The following dataset was generated:

| Author(s) | Year | Dataset title | Dataset URL | Database and Identifier |
|---|---|---|---|---|
| Pavlova NN, King B, Thompson CB | 2020 | Translation in amino acid-poor environments is limited by tRNAGln charging | https://www.ncbi.nlm. nih.gov/geo/query/acc. cgi?acc=GSE157276 | NCBI Gene Expression Omnibus, GSE157276 |

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
