## [Decision Letter]

**Acceptance summary:**

The results in your report are both interesting and surprising in showing that general amino acid limitation leads specifically to reduced aminoacylation of glutaminyl tRNA, with attendant reduction in bulk protein synthesis, and to a specific decrease in the abundance of transcription factors, including TATA-binding protein, harboring poly-glutamine tracts. It is noteworthy that the depletion of charged tRNA^Gln^ could be linked to consumption of limiting Gln by the conversion of Gln to Glt by the enzyme glutaminase which supplies TCA cycle intermediates. This suggests a control mechanism for down-regulating transcription in response to amino acid depletion that is centered on glutamine levels and charging of glutaminyl tRNA; and is relevant to the reduced glutamine levels in tumor microenvironments.

**Decision letter after peer review:**

Thank you for submitting your article "Translation in amino acid-poor environments is limited by tRNA^Gln^ charging" for consideration by *eLife*. Your article has been reviewed by three peer reviewers, one of whom is a member of our Board of Reviewing Editors, and the evaluation has been overseen by James Manley as the Senior Editor. The following individual involved in review of your submission has agreed to reveal their identity: David M Sabatini (Reviewer #3).

The reviewers have discussed the reviews with one another and the Reviewing Editor has drafted this decision to help you prepare a revised submission.

Summary:

This paper shows that general amino acid (AA) depletion of mammalian cells in culture leads to a specific depletion of charged tRNA^Gln^, greater than for any other tRNA (charging of all tRNAs was measured); which can be reversed by inhibiting glutaminase to prevent Gln deamination to Glt. Restoring Gln-tRNA with the glutaminase inhibitor stimulates translation, measured in several ways, which then leads to deacylation of Met tRNA. Activation of Gcn2 occurs in all of these conditions, as uncharged tRNA is present for either Gln- or Met-tRNA; whereas TORC1 activation is recovered after prolonged AA starvation only in the absence of the glutaminase inhibitor. Several polyQ-tract containing transcription factors show reduced expression during AA starvation, which can be rescued by glutaminase inhibition, and the presence of a polyQ tract in a GFP reporter conferred reduced GFP expression that was shown to be dependent on Gln limitation. This was also associated with reduced bulk transcription and an increase in frameshifting in a polyQ tract.

Some revisions of text are needed to clarify some important issues raised by reviewer 2, and additional biological replicates are required for certain experiments where it appears that only technical replicates have been provided. In addition, it is requested that you knockout or knockdown glutaminase as a control for the on-target nature of the glutaminase inhibitor you employed.

Reviewer #1:

This paper shows that general amino acid (AA) depletion of mammalian cells in culture leads to a specific depletion of charged tRNA^Gln^, greater than for any other tRNA (charging of all tRNAs was measured); which can be reversed by inhibiting glutaminase to prevent Gln deamination to Glt. Restoring Gln-tRNA with the glutaminase inhibitor stimulates translation, measured in several ways, which then leads to deacylation of Met tRNA. Activation of Gcn2 occurs in all of these conditions, as uncharged tRNA is present for either Gln- or Met-tRNA; whereas TORC1 activation is recovered after prolonged AA starvation only in the absence of the glutaminase inhibitor. Several polyQ-tract containing transcription factors show reduced expression during AA starvation, which can be rescued by glutaminase inhibition, and the presence of a polyQ tract in a GFP reporter conferred reduced GFP expression that was shown to be dependent on Gln limitation. This was also associated with reduced bulk transcription and an increase in frameshifting in a polyQ tract.

It is interesting and surprising that general amino acid limitation leads specifically to enhanced deacetylation of tRNA^Gln^, with attendant activation of Gcn2 and reduction in bulk protein synthesis that increases over the time of starvation; and also that the depletion of charged tRNA^Gln^ can be linked to consumption of the limiting Gln by glutaminase conversion to Glt (to supply TCA cycle intermediates). It is also interesting that reduced translation of a polyQ reporter was observed, in parallel with depletion of several polyQ-containing transcription factors and diminished bulk transcription, which could be rescued by the glutaminase inhibitor, which the authors ascribe to the specific reduction in charged tRNA^Gln^. This suggests a control mechanism for down-regulating transcription in response to AA depletion that is centered on Gln tRNA. Important control experiments were conducted to show that deacylation of leucyl tRNAs, leading to a comparable decrease in bulk translation compared to that given by AA starvation, does not lead to reduced levels of the polyQ-containing transcription factors. They also established that decreased expression of TBP is indeed dependent on its polyQ tract by examining a variant lacking the tract; and they showed that several GTFs lacking polyQ tracts do not show reduced expression during AA starvation.

Reviewer #2:

This is a revised manuscript that addresses the consequences of depletion of exogenous amino acids on tRNA charging and protein synthesis. The study indicates that coarse amino acid depletion selectively reduces tRNA(Gln), which is accompanied by activation of the eIF2 kinase GCN2, transient diminishment of mTORC1, and lowered global translation. One reason for tRNA(Gln) being selectively deacylated upon depletion of exogenous amino acids is that glutaminase, which beneficial for biosynthesis and cell proliferation, is suggested to exacerbate depletion of intracellular glutamine. Reduced tRNA(Gln) is suggested to contribute to reduced bulk translation and facilitate translation frameshifting of mRNAs encoding proteins with polyQ segments.

Overall, this is an interesting and significant line of investigation. The tRNA(Gln) charging nd glutaminase has some new features and the integration of the frameshifting of polyQ encoding mRNAs in response to deficient amino acids is significant to understanding the consequences of depleted tRNA charging. The full biological consequences of these events for the health of cells was not explored, but the molecular features are detailed with rigor and clarity. Overall there is reviewer enthusiasm for the manuscript. There are some concerns listed below. Addressing these concerns would add clarity to the manuscript and support some stated or implied conclusions.

Reviewer concerns:

1) A feature that is not demarcated in the manuscript is the relative contributions of GCN2/eIF2 phosphorylation and depleted tRNA(Gln) to global translational control. Applications of ISRIB suggest that GCN2/eIF2 phosphorylation does not contribute to translation of a reporter GFP or frameshifting upon depletion of amino acids, but this drug and other genetic tools, along with eIF2 phosphorylation (not measured in Figure 1A or later figures), were not integrated into the amino acid deficiency model defined in Figure 1. The text is opaque on this point. For example, in the Figure 7, par. 1, it is stated that the onset of GCN2 activation was mirrored by a decline in translation. This is an accurate statement, but leaves the impression that reduced translation initiation triggered by eIF2 phosphorylation is a major contributor to lowered bulk translation. However, the GFP reporter in Figure 3D suggests otherwise.

2) Some of the figures are confusing as presented. Figure 3B, Figure 3C, Figure 4A, both panels of Figure 4B, and others indicated +AA and -AA with lines below the bar graphs. These lines are not aligned appropriately between the lanes and do not appear to be of proper length. As a consequence the lane information is not clear. Furthermore, the statistical analyses are confusing. Many of the figures are multiple technical replicates, (e.g. protein synthesis) and although there are biological replicates (sometimes only 2), the statistical analysis (error bars and p values) appears to be derived only from technical replicates from a single biological experiment.

3) Figure 2F: Explain the disconnect between ATF4 expression and maximum activation of GCN2 as measured by GCN2 phosphorylation. There is a balance between eIF2 phosphorylation and preferential translation and the available of charged tRNA to sustain translation. The text explaining ATF4 expression in Figure 2F is less than clear and in part does not appear to be accurate.

4) The text indicates that GLS expression is beneficial for fueling biosynthesis and cell proliferation when amino acids are readily availability. Is the regulation of glutaminase expression/activity during these nutrient conditions?

Reviewer #3:

Pavlova et al. unveil a novel mechanism of nutrient sensing, specifically by toggling the levels of amino acids and measuring tRNA charging with the CHARGE-seq method. They find that deprivation of amino acids specifically leads to the accumulation of the uncharged tRNA-gln while preserving the charged status of the other amino acids. The authors demonstrate that this tRNA charging is dependent on lysosomal function and accumulation of uncharged tRNA^gln^ is sufficient to activate GCN2. Consequently, the depletion of charged tRNA^gln^ causes stalling in the translation of proteins containing polyglutamine tracts such as core binding factor α1, mediator subunit 12, transcriptional coactivator CBP and TATA-box binding protein. From these results, the authors conclude that the translation of polyglutamine-encoding transcripts is an amino acid-sensing mechanism used by mammalian cells.

This work warrants publication in *eLife* and is suitable for publication as it stands. The experiments performed were rigorous, thoughtful, and well controlled. The conclusions made by the authors match the data they present. The manuscript is clearly written and the experiments are explained with sufficient detail.

The concept that amino acid deprivation causes the selective uncharging of the glutamyl-tRNA, and not other tRNAs, is very interesting. Additionally, the authors develop the CHARGE-seq method to measure tRNA charging status, which is a clever way to readout tRNA charging and will be widely useful to other scientists in the field. Finally, the fact that translation of glutamine-rich sequences is a mechanism of nutrient sensing for amino acids is novel and of interest to the broader research community.

Major comment:

Throughout the manuscript, the authors use CB-839 to inhibit glutaminase. It would strengthen the claims of the manuscript if the authors also included knockout or knockdown of Glutaminase as a control for the on-target nature of this small molecule in a few key experiments (for example, Figure 4B).

---

## [Author Response]

Revisions for this paper:Some revisions of text are needed to clarify some important issues raised by reviewer 2, and additional biological replicates are required for certain experiments where it appears that only technical replicates have been provided. In addition, it is requested that you knockout or knockdown glutaminase as a control for the on-target nature of the glutaminase inhibitor you employed.Reviewer #2:[…]1) A feature that is not demarcated in the manuscript is the relative contributions of GCN2/eIF2 phosphorylation and depleted tRNA(Gln) to global translational control. Applications of ISRIB suggest that GCN2/eIF2 phosphorylation does not contribute to translation of a reporter GFP or frameshifting upon depletion of amino acids, but this drug and other genetic tools, along with eIF2 phosphorylation (not measured in Figure 1A or later figures), were not integrated into the amino acid deficiency model defined in Figure 1. The text is opaque on this point. For example, in the Figure 7, par. 1, it is stated that the onset of GCN2 activation was mirrored by a decline in translation. This is an accurate statement, but leaves the impression that reduced translation initiation triggered by eIF2 phosphorylation is a major contributor to lowered bulk translation. However, the GFP reporter in Figure 3D suggests otherwise.

Following reviewer #2’s recommendation, we took steps to better understand the relative contribution of GCN2-driven eIF2α phosphorylation to the amino acid withdrawal-induced suppression of translation, which we now report in revised Figure 1B. To examine the relative contribution of GCN2-induced phosphorylation of eIF2α to the observed decrease in translation, we used a small molecule ISRIB, which acts by counteracting the inhibitory effect of phosphorylated eIF2α on eIF2B, the guanine exchange factor for the eIF2 translation initiation complex. Despite ISRIB readily suppressing ATF4 accumulation and eliciting a further increase in eIF2α phosphorylation in cells deprived of amino acids for 6 hours (new Figure 1—figure supplement 1B), it had only a minor effect on bulk translation at the 6-hr as well as at the 1-hr time point post-amino acid withdrawal (new Figure 1—figure supplement 1A). This result lends further support to our premise that it is primarily the depletion of substrates (i.e., charged tRNA^Gln^, as we show later in the manuscript), rather than eIF2α-mediated regulation that is the major factor that suppresses translation in cells deprived of all amino acids. We are thankful to the reviewer #2 for suggesting this experiment.

2) Some of the figures are confusing as presented. Figure 3B, Figure 3C, Figure 4A, both panels of Figure 4B, and others indicated +AA and -AA with lines below the bar graphs. These lines are not aligned appropriately between the lanes and do not appear to be of proper length. As a consequence the lane information is not clear. Furthermore, the statistical analyses are confusing. Many of the figures are multiple technical replicates, (e.g. protein synthesis) and although there are biological replicates (sometimes only 2), the statistical analysis (error bars and p values) appears to be derived only from technical replicates from a single biological experiment.

Following the reviewer #2’s recommendation, we redesigned the labeling on the figures to improve their clarity as well as re-plotted the data from O-propargyl-puromycin assays, reporter assays, cell growth experiments, etc., as averages from independent experiments (with diamonds representing data points from independent experiments), as opposed to averages from experimental replicates from a single experiment. We also recalculated the statistics accordingly, stated the new P values in the figure panels, and updated the source data. We thank the reviewer for these suggestions to enhance the clarity of the figures and strengthen the conclusions that can be drawn from the data.

3) Figure 2F: Explain the disconnect between ATF4 expression and maximum activation of GCN2 as measured by GCN2 phosphorylation. There is a balance between eIF2 phosphorylation and preferential translation and the available of charged tRNA to sustain translation. The text explaining ATF4 expression in Figure 2F is less than clear and in part does not appear to be accurate.

We followed the reviewer #2’s request for clarification regarding our interpretation of the increased accumulation of ATF4 in amino acid-poor cells treated with glutaminase inhibitors. We thank the reviewer for suggesting this additional clarification.

4) The text indicates that GLS expression is beneficial for fueling biosynthesis and cell proliferation when amino acids are readily availability. Is the regulation of glutaminase expression/activity during these nutrient conditions?

We also assessed the levels of GLS in amino acid-replete vs. amino acid-poor conditions. We have incorporated the data in the new Figure 2—figure supplement 1J, with the accompanying discussion.

Reviewer #3:Major comment:Throughout the manuscript, the authors use CB-839 to inhibit glutaminase. It would strengthen the claims of the manuscript if the authors also included knockout or knockdown of Glutaminase as a control for the on-target nature of this small molecule in a few key experiments (for example, Figure 4B).

Following Dr. Sabatini’s recommendation, we took steps to validate the on-target nature of the effect of glutaminase inhibitor CB-839 on the maintenance of polyQ protein levels and the augmentation of RNA synthesis in amino acid-depleted cells. To this end, we depleted GLS from cells via RNA interference, and observed that two shRNAs targeting GLS (both KGA and GAC isoforms) resulted in an increased level of several polyQ proteins as well as increased labeled uridine incorporation in cells cultured in amino acid-poor medium. We have included the new data in the new Figure 4—figure supplement 1D and 1E, with accompanying discussion in subsection “Polyglutamine tract-containing proteins are sensitive to amino acid depletion”.